# Consistent Diffusion Language Models

**Hasan Amin** [* 1]  **Yuan Gao** [* 2]  **Yaser Souri** [2]  **Subhojit Som** [2]  **Ming Yin** [1]  **Rajiv Khanna** [1]  **Xia Song** [2]

## Abstract

Diffusion language models (DLMs) are an attractive alternative to autoregressive models because they promise sublinear-time, parallel generation, yet practical gains remain elusive as high-quality samples still demand hundreds of refinement steps. In continuous domains, consistency training along the probability-flow ODE is a popular recipe to accelerate diffusion. For discrete diffusion, no analogous sample-space ODE exists, making direct adaptation ill-defined. We argue that the right discrete substitute is the exact posterior bridge—the closed-form conditional law linking any two noise levels—which is available for broad corruptions including masked and uniform diffusion. Building on this observation, we introduce Multi-Path Discrete Consistency (MPDC), a new principle that trains a denoiser to be path-invariant in expectation across these stochastic bridges, and instantiate it as the Consistent Diffusion Language Model (CDLM), a single-stage training framework that does not require an already trained teacher model. Our CDLM objective recovers masked diffusion, continuous consistency models, and progressive or discrete distillation as analytic limits or empirical approximations of one common view. Empirically, CDLM establishes a new state of the art on both conditional and unconditional text-generation, consistently outperforming strong base discrete diffusion models and often even multi-stage distilled baselines across sampling budgets, with the largest gains in the few-step regime. Together, these results position CDLM as a principled and scalable foundation for the next generation of fast, high-fidelity discrete generative modeling.

*Equal contribution. Work done during an internship at Microsoft. [1]Department of Computer Science, Purdue University [2]Microsoft. Correspondence to: Hasan Amin <hasanamin@purdue.edu>.

*Proceedings of the $43^{rd}$ International Conference on Machine Learning*, Seoul, South Korea. PMLR 306, 2026. Copyright 2026 by the author(s).

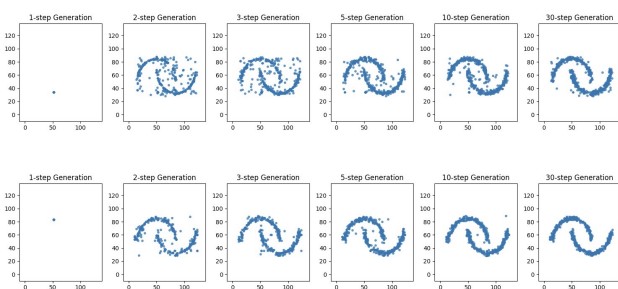

*Figure 1.* **Illustrative toy example on 2D moons under discrete diffusion.** The continuous moons data are quantized into tokens and modeled as a language-like sequence. Standard masked diffusion (top) forms sharp structure only after 10+ denoising steps, while CDLM (bottom) yields clear samples within 2–3 steps and continues to improve with larger budgets.

## 1. Introduction

Diffusion models have emerged as a dominant paradigm in generative modeling, achieving state-of-the-art results across continuous domains such as images, audio, and video (Yang et al., 2023). Their appeal lies in the simple principle of *iterative refinement*: data are gradually corrupted into noise and then reconstructed step by step. This formulation has proven both scalable and versatile.

Recently, the diffusion paradigm has extended to language, where its promise lies not just in quality but in offering a fundamentally different path to scalable generation (Austin et al., 2021). Unlike autoregressive (AR) models constrained to sequential, left-to-right decoding, diffusion language models treat generation as iterative denoising and can update many token positions in parallel, offering a route to sublinear-time sampling in sequence length (Li et al., 2022). Among these, masked diffusion language models (MDLMs) have shown strong empirical results, rivaling AR baselines (Sahoo et al., 2024; Nie et al., 2025). However, the potential of DLMs remains largely untapped, as high-quality generation typically requires hundreds of refinement steps, eroding the efficiency gains the formulation was meant to deliver. Accelerating DLMs without sacrificing quality is now a central open challenge.

In continuous domains, consistency models meet this challenge by training a network whose prediction is invariant along a single denoising trajectory (Song et al., 2023). That

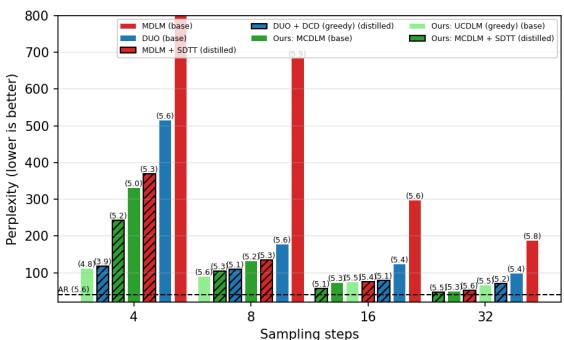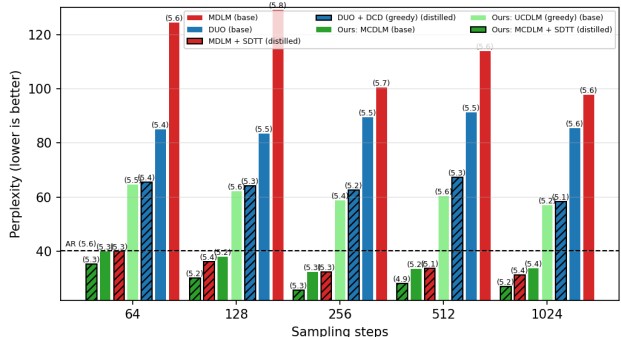

*Figure 2.* Unconditional generative perplexity (entropy in parentheses) vs. FP64 sampling steps on OpenWebText. Hatched bars (▧) denote distilled models, while colors distinguish MDLM, DUO, and CDLM families (MCDLM and UCDLM denote CDLM with masked and uniform corruption, respectively). CDLM offers the best single-stage model across budgets and often matches distilled baselines with healthier entropy, and incorporating explicit distillation leads to further improvements. See Section 4.3 for more details.

trajectory is defined by the *probability-flow ordinary differential equation* (PF-ODE). For any noisy sample $x_t$, the ODE specifies a unique deterministic path linking $x_t$ to the clean data $x_0$. Consistency training then matches the network's outputs at different times along this same path, which is what enables few- or even one-step generation after training. *Discrete categorical diffusion admits no analogous sample-space ODE.* Corrupting or denoising a token sequence is inherently stochastic, as many distinct predecessor sequences can lead to the same noisy state. Thus there is no unique path on which to enforce pointwise consistency, and direct imports of continuous consistency training are ill-defined without surrogates or multi-stage distillation (Deschenaux & Gulcehre, 2025; Sahoo et al., 2025).

We close this gap with *Multi-Path Discrete Consistency (MPDC)*, a discrete analogue of consistency training. Even though a unique deterministic path is absent, discrete diffusion already supplies an *exact posterior bridge*—the conditional law $q(x_s \mid x_t, x_0)$ connecting any two noise levels $s < t$—in closed form for broad corruptions, including masked and uniform diffusion (Austin et al., 2021). A denoising *path* is then any composition of such bridges (a direct $t \to s$ jump or a longer chain). MPDC requires a time-conditional predictor to agree across these paths *in expectation*, replacing ODE path-invariance with bridge path-invariance. Few-step generation emerges not as an approximation, but a consequence of such training: once long and short routes are equivalent in expectation, the model can *choose to* take large denoising jumps at inference without relying on a teacher rollout.

Instantiating this principle, we propose the *Consistent Diffusion Language Model (CDLM)*. CDLM trains a predictor $f_\theta(x_t, t)$ so that its output at a noisier state $(x_t, t)$ matches the output at a cleaner state $(x_s, s)$ obtained by first sampling $x_s \sim q(x_s \mid x_t, x_0)$ on the true bridge—enforcing that denoising from $x_t$ is consistent with hopping to $x_s$ first,

then denoising. We focus on masked text diffusion, but the recipe applies whenever posterior bridges are tractable. Training on bridges of mixed length teaches the model to denoise across large time gaps, which is what makes high-quality generation possible in only a handful of steps.

The MPDC objective also unifies several acceleration paradigms as special or approximate cases of one bridge-consistency view. Standard masked diffusion (Sahoo et al., 2024; Shi et al., 2024) corresponds to a max-step bridge anchor; continuous consistency models (Song et al., 2023) arise as the deterministic limit; progressive distillation (Salimans & Ho, 2022) and shortcut models (Frans et al., 2025) replace analytic bridges with rolled-out trajectories; and recent discrete distillation methods (Deschenaux & Gulcehre, 2025; Sahoo et al., 2025) implement teacher-defined or comonotonic pseudo-bridges. Within this taxonomy, CDLM is the teacher-free, single-stage member that supervises from the exact posterior bridge supplied by the discrete process itself, rather than from a learned teacher trajectory or continuous surrogate. Empirically (Figure 2), CDLM sets a new best among single-stage discrete diffusion models across sampling budgets and often matches or beats multi-stage distilled baselines while retaining healthier sample diversity.

Our contributions are threefold:

1. *A new principle for discrete generative modeling.* We introduce multi-path discrete consistency and show that exact posterior bridges are a rigorous, closed-form substitute for the absent PF-ODE, turning the main obstacle to discrete acceleration into an analytic asset.

2. *A unified and general training framework.* We present a self-contained CDLM objective for training path-invariant discrete denoisers, applicable across corruption processes, and show that standard diffusion, consistency, and distillation-like objectives are special cases or approximations of it.

3. *State-of-the-art text generation, without a teacher.* On standard benchmarks, CDLM outperforms base DLMs and matches or surpasses optimized multi-stage distilled models across sampling budgets, despite being trained in a single stage from scratch. A distilled variant pushes performance further, achieving up to $32\times$ speedups over autoregressive baselines.

CDLM reframes fast discrete diffusion as the problem of learning a *path-independent denoiser*, and shows that doing so gives a single-stage, teacher-free model that already operates at the frontier of discrete diffusion acceleration. We hope this perspective serves as a principled foundation for scalable, high-fidelity discrete generative modeling.

## 2. Problem Setup

We ground our framework in the standard formalism of discrete diffusion (Austin et al., 2021), which defines a forward-time corruption process that gradually transforms data into noise, together with a parameterized reverse process that reconstructs data from noise. In discrete domains such as text, states are sequences of categorical variables. To avoid ambiguity between sequence-level and token-level operations, let $\boldsymbol{x} \in \mathcal{V}^L$ denote a sequence of length $L$ over a vocabulary $\mathcal{V}$. The state at time $t$, $\boldsymbol{x}_t = (x_t^1, \ldots, x_t^L)$, consists of discrete tokens $x_t^i \in \mathcal{V}$. The forward process is a non-homogeneous Markov chain that factorizes independently over token positions. For a single token position, the transition is governed by matrices $\boldsymbol{Q}_t \in \mathbb{R}^{|\mathcal{V}| \times |\mathcal{V}|}$:

$$q(\boldsymbol{x}_t \mid \boldsymbol{x}_0) = \prod_{i=1}^{L} q(x_t^i \mid x_0^i) = \prod_{i=1}^{L} \text{Cat}(x_t^i; \ x_0^i \boldsymbol{Q}_{1:t}),$$

with $\boldsymbol{Q}_{a:b} \coloneqq \prod_{s=a}^{b} \boldsymbol{Q}_s$ where $\boldsymbol{x}_0^i$ denotes the one-hot row vector of the token at position $i$.

Each $\boldsymbol{Q}_t$ is row-stochastic to conserve probability mass. Additionally, rows of $\boldsymbol{Q}_{1:t}$ must converge to a known stationary distribution over time, ensuring that $q(\boldsymbol{x}_t)$ approaches a tractable prior over time. Using $\langle \cdot, \cdot \rangle$ for Euclidean inner product and $\odot$ for elementwise product, the exact posterior at time $t-1$ for a single token $x^i$ is written in closed form:

$$q(x_{t-1}^i \mid x_t^i, x_0^i) = \text{Cat}\left(x_{t-1}^i; \frac{x_t^i \boldsymbol{Q}_t^\top \odot \ x_0^i \boldsymbol{Q}_{1:t-1}}{\langle x_0^i \boldsymbol{Q}_{1:t}, x_t^i \rangle}\right).$$

A particularly important instance for language is *masked (or absorbing-state) diffusion*, where the stationary distribution places all probability on a special [MASK] token. Not only is masking found to be the most effective corruption (Austin et al., 2021; Sahoo et al., 2024; Shi et al., 2024; Nie et al., 2025), but it also helps simplify the closed-form marginals and posteriors. Masked diffusion language models exploit this to parameterize $\boldsymbol{x}_0$ directly, enabling efficient sampling.

**Continuous diffusion and a natural notion of consistency.** In continuous domains, a standard construction of the forward process is a variance-exploding (VE) stochastic differential equation (SDE):

$$d\boldsymbol{x}_t = g(t) \, d\boldsymbol{w}_t, \qquad t \in [0, 1], \qquad \boldsymbol{x}_0 \sim p_{\text{data}} \quad (1)$$

where $\boldsymbol{w}_t$ is a Wiener process, $g(t) \geq 0$ is a noise schedule, and $\pi$ is a tractable stationary prior (commonly Gaussian). Equivalently, this forward diffusion implies an additive-noise perturbation view $\boldsymbol{x}_t = \boldsymbol{x}_0 + \sigma(t) \, \epsilon$, with $\epsilon \sim \mathcal{N}(0, \boldsymbol{I})$, $\sigma^2(t) = \int_0^t g(u)^2 \, du$ and $\frac{d}{dt}\sigma^2(t) = g(t)^2$. Let $p_t(\boldsymbol{x})$ denote the density of $\boldsymbol{x}_t$. The corresponding reverse-time generative dynamics can be written as a reverse SDE with drift given by the score $\nabla_{\boldsymbol{x}} \log p_t(\boldsymbol{x})$. An equivalent deterministic formulation is given by the *probability-flow ODE*:

$$\frac{d\boldsymbol{x}_t}{dt} = -\tfrac{1}{2}g(t)^2 \nabla_{\boldsymbol{x}_t} \log p_t(\boldsymbol{x}_t) = -\dot{\sigma}(t)\sigma(t)\nabla_{\boldsymbol{x}_t} \log p_t(\boldsymbol{x}_t)$$

This ODE defines a *single deterministic trajectory* (given an initial noise sample) that transports probability mass from $p_1 \approx \pi$ back to $p_0 = p_{\text{data}}$. The PF-ODE thus ties all noise levels together, and one can enforce consistency along the path by matching predictions for all points on that path. This notion of "single-path" consistency has been found promising to developing powerful models for few-step generation in continuous domain (Song et al., 2023; Song & Dhariwal, 2024), although such training can be practically challenging (Geng et al., 2025).

**Need for a new consistency formulation in discrete space.** In discrete diffusion, different corruption levels *do not* lie on a unique trajectory. There is no PF-ODE, and hence no canonical map $\boldsymbol{x}_t \mapsto \boldsymbol{x}_s$. This absence has been the primary obstacle to developing a principled consistency framework for discrete data. Our work introduces a conceptual shift: instead of searching for a non-existent deterministic path, we leverage the rich web of *stochastic* paths. Our key observation is that the discrete diffusion framework (Austin et al., 2021) provides an analytic family of such paths connecting any two noise levels, which is a powerful yet overlooked property of these diffusion processes. CDLM replaces the missing PF-ODE with these exact posterior bridges and enforces *multi-path discrete consistency*; predictions must agree across many valid routes in expectation, so short and long denoising paths can become interchangeable.

## 3. Method

We present Consistent Diffusion Language Models (CDLM), a discrete generative framework built on the principle of Multi-Path Discrete Consistency (MPDC). Prior consistency methods in continuous domains rely on the PF-ODE to define a unique, deterministic trajectory connecting noise to

data. In discrete space, instead of attempting to discretize a non-existent trajectory, we embrace the stochastic nature of discrete diffusion. We observe that the discrete framework defines a rich family of stochastic paths connecting any two noise levels via exact posterior bridges.

We generalize consistency to this stochastic regime: we train a time-conditional predictor to be *path-independent in expectation*. This means that a direct prediction from a highly corrupted state $\boldsymbol{x}_t$ must agree (in expectation) with the prediction made after taking an intermediate "hop" to $\boldsymbol{x}_s$ via any valid stochastic bridge. By enforcing this consistency, CDLM learns to decompose the difficult mapping $\boldsymbol{x}_t \rightarrow \boldsymbol{x}_0$ into arbitrary sub-problems, enabling high-quality generation in few steps as an emergent property of training.

### 3.1. Learning a Path-Independent Denoiser

We leverage the analytic reversibility of the discrete process between *arbitrary* timesteps to learn a consistent denoiser.

**Lemma 3.1** (General Posterior Bridge; Adapted from Austin et al. (2021)). *For any $0 \leq s < t$, the analytic posterior bridge for a single token position is given by:*

$$q(x_s^i \mid x_t^i, x_0^i) = \mathrm{Cat}\left(x_s^i; \frac{(x_0^i \boldsymbol{Q}_{1:s}) \odot (x_t^i \boldsymbol{Q}_{s+1:t}^\top)}{\langle x_0^i \boldsymbol{Q}_{1:t}, x_t^i \rangle}\right) \tag{2}$$

*The sequence-level bridge is the product over all positions: $q(\boldsymbol{x}_s \mid \boldsymbol{x}_t, \boldsymbol{x}_0) = \prod_i q(x_s^i \mid x_t^i, x_0^i)$. Furthermore, these bridges compose transitively, obeying a semigroup (Chapman–Kolmogorov) property: for any $0 \leq u < s < t$, $q(x_u^i \mid x_t^i, x_0^i) = \sum_{x_s^i} q(x_u^i \mid x_s^i, x_0^i) \, q(x_s^i \mid x_t^i, x_0^i)$, i.e., traversing the two-leg bridge $t \rightarrow s \rightarrow u$ and marginalizing $x_s^i$ recovers the direct bridge $t \rightarrow u$.*

Using the bridge operator, we can define what it means for a denoising function to be consistent across different paths. We seek to learn a function $f_\theta(\boldsymbol{x}_t, t)$ that predicts clean data from any noisy input $\boldsymbol{x}_t$, factorized over positions as $f_\theta(\boldsymbol{x}_t, t) = \prod_i f_\theta(\boldsymbol{x}_t, t)_i$ with $f_\theta(\boldsymbol{x}_t, t)_i \in \Delta^{|\mathcal{V}|}$. By marginalizing out the unobservable true data $\boldsymbol{x}_0$, we define consistency strictly over the true unconditional reverse transition $p(\boldsymbol{x}_s \mid \boldsymbol{x}_t) = \mathbb{E}_{\boldsymbol{x}_0 \sim p(\boldsymbol{x}_0 \mid \boldsymbol{x}_t)}[q(\boldsymbol{x}_s \mid \boldsymbol{x}_t, \boldsymbol{x}_0)]$.

**Definition 3.2** (Multi-Path Discrete Consistency Operator). Let $g(\cdot, \cdot)_i : \mathcal{V}^L \times [0, 1] \rightarrow \Delta^{|\mathcal{V}|}$ be a per-position time-conditional predictor. The multi-path discrete consistency operator, $\mathcal{C}_{s \leftarrow t}$, transforms this function as follows:

$$\left[\mathcal{C}_{s \leftarrow t} g\right](\boldsymbol{x}_t, t)_i := \mathbb{E}_{\boldsymbol{x}_s \sim p(\boldsymbol{x}_s \mid \boldsymbol{x}_t)}\left[g(\boldsymbol{x}_s, s)_i\right]. \tag{3}$$

This operator returns the expected per-position prediction of $g$ at time $s$, after transitioning from time $t$ via the exact unconditional reverse chain.

The ideal denoising function would be a fixed point of this

operator for all possible timesteps, anchored by the true data at the boundary.

**Definition 3.3** (Global Multi-Path Consistency). A function $f^\star$ is globally multi-path-consistent if it satisfies the boundary condition $f^\star(\boldsymbol{x}_0, 0) = \boldsymbol{x}_0$ and, for all $0 < s < t \leq 1$, it is a fixed point of the expected consistency operator:

$$f^\star(\boldsymbol{x}_t, t) = \left[\mathcal{C}_{s \leftarrow t} f^\star\right](\boldsymbol{x}_t, t). \tag{4}$$

The corresponding population objective, the *expected consistency loss*, measures the discrepancy between a candidate predictor and its image under $\mathcal{C}_{s \leftarrow t}$:

$$\mathcal{L}_{\mathrm{cons}}(f) := \mathbb{E}_{t, s, \boldsymbol{x}_t}\left[\mathbb{D}\big(f(\boldsymbol{x}_t, t) \,\|\, [\mathcal{C}_{s \leftarrow t} f](\boldsymbol{x}_t, t)\big)\right], \tag{5}$$

where the expectation is over $0 < s < t \leq 1$ and $\boldsymbol{x}_t \sim p_t$, and $\mathbb{D}$ is a strictly proper divergence.

**Proposition 3.4** (Bayes fixed point). *Within the factorized class, let $f^*(\boldsymbol{x}_t, t)_i := p(x_0^i \mid \boldsymbol{x}_t)$ denote the per-position posterior marginal over clean tokens. Then $f^*$ is a fixed point of the multi-path consistency operator: $f^*(\boldsymbol{x}_t, t) = [\mathcal{C}_{s \leftarrow t} f^*](\boldsymbol{x}_t, t)$ for all $0 < s < t \leq 1$. Moreover, if the boundary edge $s = 0$ is included—equivalently, if a positive-weight max-step diffusion anchor is added under a strictly proper mean-eliciting scoring rule $\mathbb{D}$ (e.g., forward KL / cross-entropy)—then $f^*$ is the unique population minimizer within the factorized class.*

This condition formalizes path-invariance: predicting from $\boldsymbol{x}_t$ directly is equivalent to first transitioning to *any* intermediate state $\boldsymbol{x}_s$ via the true reverse chain and predicting from there. The unanchored self-consistency loss is not by itself identifying, and degenerate path-invariant predictors (e.g., constant-in-$t$ functions matching the boundary on a measure-zero set) can also be fixed points. This is also why CDLM includes the max-step diffusion anchor in Eq. 7, which grounds the consistency equations at the data boundary and selects the Bayes posterior marginal. Moreover, the optimum within the factorized class is the per-token posterior marginal, not the joint $p(\boldsymbol{x}_0 \mid \boldsymbol{x}_t)$. CDLM therefore does not aim to overcome the token-factorization barrier of one-step discrete generation from maximally corrupted noise, and targets the few-step regime, where multiple refinement passes progressively resolve cross-token structure.

**Training.** We train a model $f_\theta$ to satisfy the global consistency property by minimizing the discrepancy between the two sides of Eq. 4 over a random selection of timesteps and data. Enforcing local consistency across edges rigorously bounds global path-independence (see Appendix C). For $\delta = t - s$, the CDLM objective $\mathcal{L}_{\mathrm{CDLM}}(\theta)$ evaluates:

$$\mathbb{E}_{\boldsymbol{x}_0, t, \delta, \boldsymbol{x}_t}\left[\sum_{i \in \mathcal{M}(\boldsymbol{x}_t)} w(t, \delta) \cdot \mathbb{D}\big(f_\theta(\boldsymbol{x}_t, t)_i \,\big\|\, f_{\tilde{\theta}}(\boldsymbol{x}_s, s)_i\big)\right], \tag{6}$$

where $\boldsymbol{x}_s$ is sampled from the *oracle* bridge $q(\boldsymbol{x}_s \mid \boldsymbol{x}_t, \boldsymbol{x}_0)$ using ground-truth $\boldsymbol{x}_0$; this is a standard Monte Carlo estimator of the unconditional consistency target $p(\boldsymbol{x}_s \mid \boldsymbol{x}_t)$ in Eq. 3. The summation index set $\mathcal{M}(\boldsymbol{x}_t) \subseteq \{1, \ldots, L\}$ specifies which token positions contribute to the loss, which allows a unified formulation across different discrete corruption kernels. For absorbing-state (masking) diffusion, we set $\mathcal{M}(\boldsymbol{x}_t) = \{i : x_t^i = [\texttt{MASK}]\}$. For non-absorbing priors such as uniform diffusion, we simply use all positions, $\mathcal{M}(\boldsymbol{x}_t) = \{1, \ldots, L\}$. Here, $f_{\tilde{\theta}}$ denotes a target network whose parameters $\tilde{\theta}$ are a variant of $\theta$ (e.g., a slow-moving exponential average) to stabilize training. The term $\mathbb{D}$ is a divergence measure, and $w(t, \delta)$ is a positive weighting function. Precise algorithmic formulations for both a general training recipe (Consistent Discrete Denoising Diffusion Training; CD3T) and a concrete instantiation within masked diffusion context (M-CDLM) are deferred to Appendix B.

**Sampling.** A trained CDLM is a time-conditional denoiser, analogous to a standard MDLM, which allows it to leverage existing sampling. We use ancestral sampling, where given a sequence $\boldsymbol{x}_t$, we first predict its clean version $\hat{\boldsymbol{x}}_0 = f_\theta(\boldsymbol{x}_t, t)$ and then use the posterior bridge $q(\boldsymbol{x}_s \mid \boldsymbol{x}_t, \hat{\boldsymbol{x}}_0)$ to sample the next state $\boldsymbol{x}_s$ (Austin et al., 2021; Sahoo et al., 2024). The number of steps in this iterative process is a flexible hyperparameter at inference time and is decoupled from the training formulation. Note that CDLM's novelty does *not* lie in devising new samplers, but in training a model that remains robust under any schedule of steps, although compatibility with existing samplers helps with fair comparison and adoption.

### 3.2. Design Insights for Stable and Scalable Training

While the default CDLM objective in Eq. 6 suffices for simple settings, such as the 2D moons data in Fig. 1, the MPDC objective is self-referential, creating an optimization landscape where it takes very long to converge or leads to degenerate solutions. In particular, naive optimization could lead to *mode collapse*, where the outputs become overly repetitive and deterministic to trivially satisfy consistency, or *uniform drift*, where predictions degrade towards uninformative distributions that are easy to 'match'. We introduce three principled design choices that stabilize training and scale CDLM effectively in practice.

**Step size as a multi-task curriculum.** In CDLM, the step size $\delta = t - s$ determines how far we "jump" along a denoising route. Through linearity of expectation, we can view CDLM training as *multi-task learning over step sizes*: each $\delta$ specifies a distinct path-equivalence constraint. This perspective makes two design questions explicit: (i) *which* step sizes should be practiced (the step size scheduler $p(\delta)$), and (ii) *how* to weight them (the weighting scheduler $w(\delta)$).

We sample $\delta$ within a practical range (e.g., $1/8$–$3/8$), which directly targets the few-step regime. Moreover, we select $w(\delta) = \frac{1}{\delta}$ to help with *path length normalization*. If $\delta$ is sampled uniformly, longer segments are rarer but cover more 'time volume,' while shorter segments are frequent but cover less. Our choice of $\frac{1}{\delta}$ ensures that, in expectation, each unit of 'corruption time' contributes equally to the total loss, balancing the learning signal across short and long paths In practice this choice prevents the training signal from being dominated by easy, local constraints while still supplying dense supervision where it is most stable.

**A diffusion anchor via max-step scheduler.** The self-referential nature of the CDLM loss can be stabilized by grounding it with the true data distribution. We mix in a small fraction of "max-step" tasks where $\delta = t$ (so $s = 0$). The corresponding loss $\mathcal{L}_{\text{final}}(\theta)$ then becomes:

$$(1 - \kappa_{\text{ms}}) \mathcal{L}_{\text{CDLM}}(\theta) + \kappa_{\text{ms}} \mathbb{E}_{t, \boldsymbol{x}_0, \boldsymbol{x}_t} \left[ \frac{1}{t} \mathbb{D}(f_\theta(\boldsymbol{x}_t, t) \| \boldsymbol{x}_0) \right],$$
(7)

which recovers the standard diffusion objective as a regularizer. In practice, a small $\kappa_{\text{ms}} \in [0.1, 0.4]$ suffices to ground learning and discourage low–entropy "shortcut" solutions. Moreover, we find this regularization is most critical in early stages of training and its weight can be annealed over time.

**Optimization asymmetry and choice of divergence.** To prevent the model from collapsing by perfectly matching its own (potentially flawed) predictions, we introduce an optimization asymmetry. This is implemented using a stop-gradient on the target network, whose parameters are a slow-moving exponential average (EMA) of the online model (Grill et al., 2020). Furthermore, to balance the mode-seeking and mode-covering tendencies of forward and reverse KL-divergence, which can exacerbate collapse and drift respectively, we use the symmetric and bounded Jensen-Shannon Divergence, which provides more stable gradient signal when training from scratch.

### 3.3. A Unifying View of Discrete Generative Modeling

The MPDC principle not only enables efficient generation but also provides a general lens through which to understand and connect a range of modern generative models. We now show that the canonical objectives for masked diffusion, consistency models, and other acceleration techniques emerge as specific instantiations of the CDLM framework. More rigorous equivalences are shared in Appendix D.

**Masked Diffusion Models (Sahoo et al., 2024) as the max-step boundary.** Under the maximum step size $\delta = t$, the target collapses to the deterministic clean data. Configuring CDLM with Forward KL and the continuous-time weight

$w(t) = -\frac{\alpha_t'}{1-\alpha_t} = 1/t = 1/\delta$ mathematically recovers the exact continuous-time NELBO optimized by MDLM.

**Consistency Models (Song et al., 2023) as the deterministic continuous analog.** Continuous Consistency Models enforce local self-consistency across adjacent steps by relying on an ODE solver to deterministically couple points. CDLM generalizes this to discrete Markov spaces by replacing the non-existent deterministic ODE step with an expectation over the exact stochastic posterior bridge.

**Analytical bypass of empirical bootstrapping (Progressive Distillation (Salimans & Ho, 2022) and Shortcut Models (Frans et al., 2025)).** Methods like Progressive Distillation (PD) and Shortcut Models accelerate continuous generation by training a model to match a multi-step trajectory using empirical bootstrapping. Because deterministic PF-ODE solvers lack analytic integrals for finite step sizes, these methods must explicitly instantiate and sum over intermediate states via neural rollouts. However, in discrete space, the true diffusion transition matrices inherently obey the Chapman-Kolmogorov equation. The exact posterior bridge analytically marginalizes over the intermediate state in closed form. Training CDLM over step size $\delta$ therefore evaluates the exact marginalized multi-step bridge in one stage, aligning with the *structural* multi-step constraints targeted by PD and Shortcut Models, but without neural rollouts or recursive bootstrapping.

**Discrete distillation as approximate bridge implementations.** CDLM is a *single-stage* framework that derives supervision directly from the *exact analytic* bridge. Recent *two-stage* distillation methods formally operate by substituting the exact bridge with *approximate empirical couplings*:

- **Self-Distillation Through Time (SDTT)** (Deschenaux & Gulcehre, 2025) constructs targets via multi-step teacher rollouts. This effectively substitutes the oracle data $\boldsymbol{x}_0$ in the analytic bridge with an empirical prediction $\hat{\boldsymbol{x}}_0 \sim p_{\text{teacher}}(\cdot \mid \boldsymbol{x}_t)$. CDLM can thus be viewed as the oracle-teacher limit of SDTT.

- **DUO-DCD** (Sahoo et al., 2025) exploits "diffusion duality" to map continuous ODE states to the discrete domain via an `argmax` projection. By sharing continuous Gaussian noise $\epsilon$ across timesteps to form a Deterministic Discrete Trajectory, it constructs a highly correlated *comonotonic pseudo-bridge* that differs structurally from the conditionally independent categorical posterior bridge used by CDLM. We hypothesize that this difference contributes to the lower-entropy behavior empirically observed for greedy DUO-DCD samples, whereas CDLM's exact bridge preserves diversity by construction.

# 4. Experiments

We evaluate the CDLM framework on both unconditional and conditional text generation. Our experiments are designed to demonstrate that CDLM not only establishes a new state-of-the-art for base, single-stage discrete diffusion models but also rivals or outperforms complex, multi-stage distilled models across various sampling budgets.

## 4.1. Related Baselines and Model Categorization

We benchmark our framework against the current state-of-the-art in discrete diffusion language modeling: MDLM (Sahoo et al., 2024), SDTT (Deschenaux & Gulcehre, 2025), and DUO (including DUO-DCD) (Sahoo et al., 2025). We briefly introduce the baseline models here, and defer a more elaborate overview of other related works to Appendix A.

Conceptually, the landscape of efficient discrete diffusion models is divided into two distinct classifications:

- **Base Models (Trained from Scratch):** Models trained in a single stage using a primary objective. MDLM is a text-based diffusion model with a masked prior trained via the NELBO loss. DUO improves upon the original Uniform Diffusion Language Models (UDLM) (Schiff et al., 2024) by leveraging a connection to continuous Gaussian distributions through an `argmax` operation. Like MDLM and DUO, our CDLM is trained purely with consistency loss from scratch.

- **Distilled Models (Multi-Stage):** Models relying on a pre-trained base model as a teacher and requiring single or multiple steps of teacher roll-outs for better generation quality across different sampling steps. SDTT performs self-distillation based on MDLM. DUO-DCD applies consistency distillation over DUO's continuous proxy, finding that a greedy sampler further improves sampling metrics.

CDLM belongs to the first category, yet our experiments demonstrate that its native multi-path discrete consistency formulation allows it to rival or surpass multi-stage models.

### 4.1.1. UNCONDITIONAL GENERATION

## 4.2. Experimental Setup

We present two primary 110M-parameter models trained with a masked source distribution: **MCDLM** (Masked CDLM) and **MCDLM-PPLOptimized**. Both models are trained within a single stage for 150K steps using Algorithm 2 with the multi-scheduler objective from Equation 7. MCDLM-PPLOptimized is a CDLM variant tuned to significantly improve generative perplexity while slightly sacrificing entropy, highlighting our framework's engineer-

| Model | Pretrain Steps | Distill Steps | Sampling steps with FP64 Sampling | | | | | | | | |
|---|---|---|---|---|---|---|---|---|---|---|---|
| | | | 4 | 8 | 16 | 32 | 64 | 128 | 256 | 512 | 1024 |
| *Comparison with Base Models (Trained from Scratch)* | | | | | | | | | | | |
| AR | 75K | 0 | N/A | N/A | N/A | N/A | N/A | N/A | N/A | N/A | 40.2 (5.6) |
| MDLM | 150k | 0 | 1654.5 (5.8) | 682.7 (5.9) | 297.1 (5.6) | 186.9 (5.8) | 124.4 (5.6) | 129.2 (5.8) | 100.5 (5.7) | 114.0 (5.6) | 97.7 (5.6) |
| Ours: MCDLM | 150k | 0 | 649.4 (5.5) | 246.9 (5.6) | 125.6 (5.4) | 86.5 (5.6) | 67.7 (5.6) | 66.0 (5.5) | 55.4 (5.5) | 58.4 (5.4) | 53.4 (5.5) |
| Ours: MCDLM–PPLOptimized | 150k | 0 | 331.2 (5.0) | 132.1 (5.2) | 71.6 (5.3) | 48.7 (5.3) | 40.1 (5.3) | 38.1 (5.2) | 32.5 (5.3) | 33.5 (5.2) | 33.8 (5.4) |
| *Comparison with Distilled Models* | | | | | | | | | | | |
| MDLM - SDTT | 100k | 50k | 369.6 (5.3) | 134.0 (5.3) | 76.0 (5.4) | 51.4 (5.6) | 40.1 (5.3) | 36.2 (5.4) | 32.5 (5.3) | 33.8 (5.1) | 31.2 (5.4) |
| Ours: MCDLM + SDTT | 100k | 50k | **242.1 (5.2)** | **105.0 (5.3)** | **57.5 (5.1)** | **47.0 (5.5)** | 35.3 (5.3) | 30.3 (5.2) | **25.8 (5.3)** | **28.1 (4.9)***  | 27.1 (5.2) |

*Table 1.* Generative perplexity (with entropy in parentheses) across different models with *Masked Distribution* as prior which we call MCDLM, training setups, and FP64 sampling steps. We use ancestral sampler for all models. Results with best PPLs are **bolded** and second best are underlined. * denotes the entropy is lower than 5 which we found empirically yield repetitive characters. Consistent with MDLM, our AR baseline is trained with half of the steps to ensure the number of total seen tokens are the same during training.

ability. We compare our models against both base and distilled baselines on unconditional and conditional generation.

For MCDLM, we set our step-size scheduler $\Delta_T$ to sample randomly in $[\frac{1}{8}, \frac{5}{8}]$, with the max-step regularizer weight $\kappa_{ms}$ set to $0.4$. For MCDLM-PPLOptimized, we use the identical setting for the first 100K steps, before gradually shrinking the maximum range of $\Delta_T$ to $\frac{3}{8}$ and annealing $\kappa_{ms}$ to $0.2$. To stabilize training (Sahoo et al., 2025; Schiff et al., 2024; Song et al., 2023), we maintain an Exponential Moving Average ($\lambda = 0.999$) for the target network $\bar{\theta}$ during CDLM training, switching to a hard update every 10K steps starting at 100K steps for the PPLOptimized variant.

To show the generalizability of our algorithm, we also conduct preliminary experiments using a model trained with a Uniform Distribution prior (**UCDLM**), while keeping the same data setup. For training, we use a linearly increasing step-size scheduler from $0.125$ to $0.375$. The rest of the configuration is kept aligned with MCDLM.

Consistent with our models, we train all compared baselines at the 110M parameter scale for 150K steps with a batch size of 2048 using OpenWebText (Gokaslan & Cohen, 2019) pretraining data. MDLM was trained with the NELBO objective for 150K steps, and SDTT undergoes a pretraining stage of 100K steps using MDLM's objective before shifting to distillation with 2 teacher updates per step for 50K steps. For DUO, similar to Sahoo et al. (2025), we always use half of the steps for curriculum learning and half of the steps for continual finetuning. DUO+DCD leverages the DUO trained with 100K steps and then uses an additional 50K steps for distillation with an updating teacher and doubling the step size $\delta$ for every 10K rounds.

### 4.3. Results and Analysis

We present results for unconditional generation with 1024 tokens in Figure 2 and Table 1 for 64-bit sampling, and Table 7 for 32-bit sampling. Each model generates 32 samples for PPL evaluation under `gpt2-large`.

**Superiority over base models.** Our MCDLM model outperforms MDLM across all sampling steps, regardless of sampling precision. Compared to DUO, MCDLM produces much lower PPLs from step 16 to 1024 under both 32- and 64-bit sampling, while maintaining a similar entropy as DUO under the 64-bit sampler.

**Rivaling multi-stage distillation.** Remarkably, despite lacking a distillation stage, our single-stage MCDLM-PPLOptimized outperforms multi-stage models like SDTT and DUO-DCD (with both ancestral and greedy samplers) across most sampling steps, while preserving a healthy entropy level. Note that although DUO-DCD with a greedy sampler yields lower PPL at very low sampling steps, it suffers from significantly lower entropy ($\sim 3.9$), which indicates severe mode collapse that produces repetitive characters and inflates the metric. We also distilled MCDLM using the SDTT objective, which yields a model strictly outperforming the original SDTT throughout steps 4 to 1024. Overall, MCDLM-PPLOptimized achieves the best balance between generative PPL and entropy.

**Speedups and efficiency.** MCDLM-PPLOptimized achieves a $64\times$–$128\times$ **speedup** over MDLM, and matches the Autoregressive (AR) baseline at 32–64 steps, a $16\times$–$32\times$ **reduction** in Number of Function Evaluations (NFE). We report NFE rather than wall-clock latency, which depends on implementation details (sequence length, batching, KV-cache support, hardware) that differ between AR and diffusion decoders. System optimizations such as diffusion KV caching and parallel decoding should be complementary to the algorithmic gains reported here.

**Uniform prior.** Finally, we observe similar success when applying Algorithm 1 to uniform prior DLMs. As shown in Table 2, our UCDLM model outperforms vanilla UDLM across all sampling steps. Moreover, compared to the state-of-the-art DUO (Sahoo et al., 2025) model and its distillation variant, UCDLM yields strictly better perplexity regardless of the sampler choice (ancestral or greedy).

| Model | Pretrain | Distill | Sampling steps with FP64 Sampling | | | | | | | | |
|---|---|---|---|---|---|---|---|---|---|---|---|
| | Steps | Steps | 4 | 8 | 16 | 32 | 64 | 128 | 256 | 512 | 1024 |
| *Comparison with Base Models (Trained from Scratch)* | | | | | | | | | | | |
| AR | 75K | 0 | N/A | N/A | N/A | N/A | N/A | N/A | N/A | N/A | 40.2 (5.6) |
| UDLM | 150k | 0 | 516.6 (5.5) | 185.9 (5.7) | 122.4 (5.6) | 93.9 (5.7) | 87.6 (5.6) | 90.5 (5.7) | 78.2 (5.7) | 83.1 (5.4) | 84.0 (5.4) |
| DUO | 150k | 0 | 514.4 (5.6) | 177.3 (5.6) | 123.2 (5.4) | 97.7 (5.4) | 85.1 (5.4) | 83.4 (5.5) | 89.4 (5.5) | 91.2 (5.5) | 85.4 (5.6) |
| Ours: UCDLM | 150k | 0 | 377.3 (5.2) | 156.9 (5.7) | 104.7 (5.5) | 85.5 (5.5) | 81.0 (5.5) | 77.6 (5.6) | 74.6 (5.5) | 71.3 (5.4) | 71.7 (5.3) |
| Ours: UCDLM (greedy) | 150k | 0 | **110.4 (4.8)**$^*$ | **89.6 (5.6)** | **74.4 (5.5)** | **65.7 (5.5)** | **64.7 (5.5)** | **62.1 (5.6)** | **58.9 (5.4)** | **60.3 (5.6)** | **57.0 (5.2)** |
| *Comparison with Distilled Models* | | | | | | | | | | | |
| DUO + DCD | 100k | 50k | 408.3 (5.6) | 166.9 (5.6) | 118.2 (5.4) | 91.8 (5.5) | 80.2 (5.5) | 79.4 (5.5) | 77.9 (5.6) | 85.8 (5.6) | 75.6 (5.5) |
| DUO + DCD (greedy) | 100k | 50k | **118.4 (3.9)**$^*$ | 109.2 (5.1) | 79.8 (5.1) | 70.5 (5.2) | 65.5 (5.4) | 64.3 (5.3) | 62.6 (5.2) | 67.3 (5.3) | 58.5 (5.1) |

*Table 2.* Generative perplexity (with entropy in parentheses) across different models with *Uniform Distribution* as prior which we call UCDLM, and FP64 sampling steps. We use ancestral sampler for all models except DUO + DCD (greedy), which uses greedy sampler described as in Sahoo et al. (2025). Results with best PPLs are **bolded**. * denotes the entropy is lower than 5 which we found empirically yield repetitive characters. Consistent with MDLM, our AR baseline is trained with half of the steps to ensure the number of total seen tokens are the same during training.

| Model | OpenWebText | | | Lambada | | | Wikitext103 | | | PTB | | |
|---|---|---|---|---|---|---|---|---|---|---|---|---|
| | 8 Step | 64 Step | 512 Step | 8 Step | 64 Step | 512 Step | 8 Step | 64 Step | 512 Step | 8 Step | 64 Step | 512 Step |
| MDLM | 45.9 / 60.9 | 40.4 / 61.1 | 39.7 / 61.1 | 72.3 / 57.6 | 62.9 / 57.7 | 61.7 / 57.9 | 44.8 / 61.9 | 39.6 / 62.2 | 39.0 / 62.2 | 248.2 / 50.5 | 220.0 / 50.7 | 215.8 / 50.6 |
| SDTT | 34.0 / 62.4 | 30.8 / 62.5 | 30.3 / 62.5 | 51.9 / 59.3 | 46.4 / 59.3 | 45.7 / 59.5 | 33.6 / 63.5 | 30.5 / 63.7 | 30.1 / 63.7 | 141.4 / 52.3 | 125.4 / 52.4 | 124.0 / 52.3 |
| Ours: MCDLM–PPLOptimized | **31.6 / 62.8** | **28.6 / 62.9** | **27.9 / 63.1** | **43.4 / 60.1** | **38.8 / 60.1** | **38.5 / 60.3** | **30.0 / 64.2** | **27.3 / 64.4** | **26.9 / 64.5** | **122.1 / 53.2** | **107.5 / 53.3** | **106.8 / 53.3** |
| DUO-DCD (Greedy) | 40.2 / 26.7 | 32.7 / 26.6 | 32.0 / 26.7 | 27.9 / 31.2 | 31.2 / 59.3 | 21.6 / 31.0 | 44.9 / 32.3 | 35.9 / 32.5 | 34.1 / 32.2 | 62.1 / 19.5 | 45.0 / 19.3 | 41.2 / 19.2 |

*Table 3.* Conditional Generation results across different datasets. Perplexity ↓ / BLEU2 ↑ results with FP64 sampling using the ancestral sampler are reported. We choose the best performing models from unconditional generation for our comparison. Results for DUO-DCD with greedy sampler are grayed out as it produces nearly random sentences that do not preserve the input conditions, which results in very low BLEU scores.

### 4.3.1. CONDITIONAL GENERATION

We also evaluate our model on conditional generation across four popular datasets, three of which are out-of-distribution (OOD): OpenWebText (Gokaslan & Cohen, 2019), Lambada (Paperno et al., 2016), Wikitext-103 (Merity et al., 2016), and PTB (Marcus et al., 1993). We randomly sample 32 sentences of 1024 tokens from each dataset, perturb 50% of the tokens using the model's prior distribution, and use them as conditions for the model to recover the original sentences. We use the original unperturbed sentences as the reference, with PPL evaluating the fluency of the final generated sentence and BLEU score assessing whether the model successfully preserves the input conditions. We also use MAUVE (Pillutla et al., 2021) to evaluate embedding space distribution matching, though this metric is over-saturated for our task (detailed in Table 8 in the Appendix).

Table 3 outlines the comparison of CDLM against SDTT and DUO. Because of its uniform distribution formulation which allows tokens to transition arbitrarily into any other tokens during sampling, DUO is not good at preserving the conditions and often yields very low BLEU scores, generating sentences that completely differ from the given condition (grayed out in the table). Conversely, MCDLM-PPLOptimized consistently outperforms SDTT in terms of both generative perplexity and BLEU score, demonstrating its robust advantage in generating plausible, consistent, and fluent sentences under given conditions.

### 4.4. Ablations and Insights

**Choice of step size scheduler (Table 4).** Other than the max-step scheduler serving as a diffusion regularizer, we also use a separate scheduler for $t$ and $\delta$ for the CDLM training. Note that training exclusively with the diffusion regularizer reduces our model directly to MDLM. We experimented with four schedulers: random, staged increasing, linear increasing, and linear decreasing. Models trained with a linear increasing scheduler are exposed to small $\delta$ early in training and larger $\delta$ later, making the final checkpoints highly effective at larger sampling steps. Conversely, linear decreasing schedulers optimize the model specifically for few-step generation.

**Max-step scheduler and diffusion regularizer (Table 5).** As theoretically motivated in Section 3.3, anchoring the CDLM objective with the true data boundary $f(\boldsymbol{x}_0, 0) = \boldsymbol{x}_0$ acts as a principled, unbiased diffusion anchor. Empirically, we confirm this is critical for balancing generation quality and diversity. Without this max-step regularization, the self-referential consistency loss is prone to rapid mode collapse, i.e., producing highly repetitive words with severely collapsed entropy and artificially biased perplexity. As we increase the weight of the diffusion regularizer ($\kappa_{ms}$), both PPL and Entropy increase across all step counts, validating its role as a structural variance balancer between unconstrained diversity and strict path consistency.

| Model | 8 | 64 | 1024 |
|---|---|---|---|
| CDLM w. random scheduler | 110.6 / 5.3 | 32.8 / 5.3 | 19.7 / 5.2 |
| CDLM w. staged increasing scheduler | 160.8 / 5.2 | 45.1 / 5.3 | 25.3 / 5.2 |
| CDLM w. linear increasing scheduler | 117.6 / 5.5 | 37.9 / 5.4 | 17.9 / 4.9 |
| CDLM w. linear decreasing scheduler | 112.3 / 5.1 | 32.1 / 5.2 | 20.5 / 5.2 |

*Table 4.* Effect of the CDLM step-size scheduler on generative perplexity and entropy for different sampling steps.

| Model | 8 | 64 | 1024 |
|---|---|---|---|
| CDLM w. no max-step scheduler | 16.7 / 3.2 | 7.1 / 3.4 | 5.7 / 2.8 |
| CDLM w. 0.4 for max-step scheduler | 110.6 / 5.3 | 32.8 / 5.3 | 19.7 / 5.2 |
| CDLM w. 1.0 for max-step scheduler | 274.0 / 5.5 | 75.1 / 5.5 | 38.4 / 5.3 |

*Table 5.* Effect of the max-step diffusion-anchor weight $\kappa_{\mathrm{ms}}$ on generation perplexity and entropy for different sampling steps.

| Model | 8 | 64 | 1024 |
|---|---|---|---|
| CDLM w. JS divergence | 110.6 / 5.3 | 32.8 / 5.3 | 19.7 / 5.2 |
| CDLM w. Forward KL | 8e4 / 6.9 | 8e4 / 6.9 | 7e4 / 6.8 |
| CDLM w. Backward KL | 75.3 / 4.6 | 64.0 / 4.4 | 44.2 / 4.3 |

*Table 6.* Effect of the consistency divergence objective on generation perplexity and entropy for different sampling steps.

**Choice of distance metric (Table 6).** Because discrete diffusion bridges are inherently stochastic, the choice of divergence $\mathbb{D}$ strictly dictates how the model handles path variance (as discussed in Section 3.3). Our ablations confirm our theoretical claims. Forward KL (targeting the exact arithmetic mean) is mathematically unbiased but highly unstable when training from scratch. It struggles to average over the massive stochasticity of the discrete jump process, which manifests as *uniform drift*, leading to catastrophic perplexity degradation. Backward KL (targeting the geometric mean) is a strong mode-seeker. It improves PPL but aggressively penalizes variance, resulting in severe *mode collapse* (dropping entropy) and stripping generative diversity. JSD provides the best stability and quality-diversity tradeoff in our setting. We note that multi-stage methods like DUO-DCD and SDTT do not suffer from these KL instabilities purely because they rely on a pre-trained, frozen teacher network to pre-condition and collapse target variance prior to distillation. CDLM tackles this natively from scratch.

## 5. Discussion and Conclusion

We introduced the Consistent Diffusion Language Model, a framework for discrete generative modeling built around enforcing multi-path discrete consistency. By supervising with exact posterior bridges, CDLM trains a path-independent denoiser for which few-step efficiency is a property of training, not a post-hoc distillation artifact. The result is a single-stage model that advances the state of scalable, high-fidelity text generation.

**Stochastic consistency vs. ODE discretization.** A critical distinction of our work is that CDLM is not a discretization of a Probability Flow ODE, but a generalization of consistency to the stochastic regime. In continuous domains, consistency models enforce invariance along a unique deterministic trajectory, enabling one-step generation. In discrete space, no such trajectory exists, so CDLM enforces invariance *in expectation* over stochastic bridges. Consequently,

the optimal one-step predictor from pure noise is the unconditional marginal distribution, which is highly multimodal. CDLM is therefore explicitly designed to resolve global structure over a *few* steps (e.g., 4–8) rather than one, successfully trading the ill-posed task of one-step discrete generation for state-of-the-art efficiency in the few-step regime.

**A general foundation for discrete domains.** Beyond masked diffusion, the CDLM framework serves as a general recipe for discrete generative modeling. As demonstrated by our results with Uniform CDLM (UCDLM), the objective effectively accelerates diverse corruption processes and outperforms standard baselines as a pre-trained foundation for distillation. This generality suggests that the framework is not tied to language alone. Additional domains involving discrete structures with tractable posteriors, such as biological sequences, graphs, or program synthesis, can benefit from this paradigm and will be interesting future work. Additionally, CDLM can serve as a stronger base model for the next generation of discrete generative methods. Many leading acceleration techniques, such as distillation, build on pre-trained base models. We show that CDLM outperforms MDLM as such a foundation, and it can serve as a promising replacement for large-scale pretraining or post-training mechanisms with downstream benefits (Nie et al., 2026).

**The design space of multi-path discrete consistency.** CDLM should be understood not as a fixed algorithm, but as a flexible framework with a rich design space. Our implementation explores one principled configuration, yet many alternatives remain, including adaptive schedules and divergence metrics. Furthermore, because CDLM is architecture- and sampler-agnostic, it directly benefits from engineering advances such as KV-caching and optimized kernels (Ma et al., 2025; Wu et al., 2026). The rapid evolution of continuous consistency models through similar refinements (Song & Dhariwal, 2024; Geng et al., 2025) suggests that CDLM is a promising starting point with significant potential for further algorithmic and wall-clock efficiency gains.

In reframing discrete diffusion as the training of a path-independent denoiser, CDLM bridges the gap between the acceleration playbooks of continuous diffusion and the realities of discrete data, laying a foundation for fast, principled, and broadly applicable generative models.

## Impact Statement

This paper advances fast, high-fidelity discrete generative modeling. By reducing the number of denoising steps needed for high-quality text generation, CDLM can lower the inference-time compute and energy cost of diffusion language models, broadening access to efficient generation. As with other language models, faster and cheaper generation can also amplify well-known risks, including the production of misleading or low-quality content. Nonetheless, CDLM introduces no new data sources or capabilities beyond those of standard diffusion language models, and existing mitigation and content-provenance practices apply directly. We do not foresee additional societal consequences that must be specifically highlighted here.

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

# A. Related Work

**Discrete Diffusion and Flow Models.** Austin et al. (2021); Lou et al. (2024) introduced diffusion models for discrete data, followed by MDLM (Sahoo et al., 2024) which demonstrated initial success on text modeling using a masked diffusion framework trained with a NELBO objective. Our work builds directly on MDLM, leveraging its simplified time-weighted cross-entropy loss structure. Parallel to diffusion, Discrete Flow Matching (Gat et al., 2024) formulates text generation by optimizing a learned marginal velocity field, yielding a training objective similar to MDLM under the masked prior. Beyond masked priors, UDLM (Schiff et al., 2024) and DUO (Sahoo et al., 2025) introduced and improved uniform prior diffusion models, unlocking higher generation quality by leveraging the uniform transition kernel for guided training and sampling, as well as discretizing continuous Gaussian distributions. While CDLM shares the single-stage training paradigm of these base models, it distinguishes itself by enforcing discrete consistency constraints to achieve efficient, high-quality generation.

**Accelerating Diffusion Language Models.** Current acceleration efforts primarily fall into two categories. The first focuses on training-free acceleration, utilizing techniques such as KV Caching (Ma et al., 2025; Liu et al., 2025b) or alternative sampling and decoding strategies (Chen et al., 2025; Huang et al., 2026; Ben-Hamu et al., 2025; Gwak et al., 2025). The second category involves training-based distillation from a pretrained teacher. For example, DUO with Discrete Consistency Distillation (DCD) (Sahoo et al., 2025) applies a consistency loss using states sampled from a discretized Gaussian path, while SDTT (Deschenaux & Gulcehre, 2025) employs self-distillation with multiple steps of teacher rollouts. Other recent approaches like Di4C (Hayakawa et al., 2025) explore distilling discrete diffusion by leveraging dimensional correlations. As shown in the next section, methods like DUO and SDTT can be viewed as special cases of the CDLM framework, which empirically achieves better generation quality without even requiring a separate teacher model.

**Consistency Models Family.** Consistency models (Song et al., 2023) were originally proposed for continuous image generation, with subsequent works improving their training stability and performance (Song & Dhariwal, 2024; Geng et al., 2025) and extending them to multistep (Heek et al., 2024) or stochastic settings (Liu et al., 2025a). More relevant to our approach are Consistency Diffusion Bridge Models (He et al., 2024), which apply consistency training to continuous diffusion bridges, while ours leverages discrete posterior bridges. We also note the similarly named Consistent Diffusion Models (Daras et al., 2023), which address sampling drift in continuous diffusion rather than discrete acceleration. In the discrete domain, CDLM extends the consistency principle to allow training from scratch with arbitrary discrete priors (e.g., masked or uniform), generalizing prior efforts that relied on continuous-to-discrete mappings.

# B. Algorithms

We present two algorithmic formulations of the CDLM training procedure. Algorithm 1 describes the general *Consistent Discrete Denoising Diffusion Training* (CD3T) recipe, applicable to any discrete corruption process with a tractable posterior bridge. Algorithm 2 instantiates CD3T for the special case of masked (absorbing-state) diffusion, which we call M-CDLM. In this case, the posterior bridge and loss simplify due to the absorbing structure, and the divergence is set to JSD with the path-length normalization weight $w = 1/\Delta_t$. Both algorithms use exponential moving average (EMA) updates for the target network parameters. Note that the max-step regularization term from Eq. 7 is implemented by mixing in samples with $\delta = t$ according to the schedule $\kappa_{\mathrm{ms}}$, which is detailed in the experimental setup (Section 4.2).

# C. From Local to Global Consistency

The following result formalizes the intuition that enforcing local consistency provides a mathematically rigorous foundation for achieving global path-independence. By drawing $\boldsymbol{x}_0 \sim \mathcal{D}$, $\boldsymbol{x}_{\tau_k} \sim q(\boldsymbol{x}_{\tau_k}|\boldsymbol{x}_0)$, and $\boldsymbol{x}_{\tau_{k-1}} \sim q(\boldsymbol{x}_{\tau_{k-1}}|\boldsymbol{x}_{\tau_k}, \boldsymbol{x}_0)$, our training scheme perfectly samples the exact joint marginal $p(\boldsymbol{x}_{\tau_k}, \boldsymbol{x}_{\tau_{k-1}})$. We therefore define our error bounds strictly in terms of the unconditional true reverse transition, which relies only on observable states.

**Lemma C.1** (Global Consistency Bound). *Let $\mathbb{D}_{TV}(\cdot, \cdot)$ be the Total Variation distance, which acts as a valid norm on the probability simplex (satisfying the triangle inequality and joint convexity). Let the expectation operator over the unconditional true reverse chain be denoted as $E_{j|i}[\cdot] \equiv \mathbb{E}_{\boldsymbol{x}_{\tau_j} \sim p(\cdot|\boldsymbol{x}_{\tau_i})}[\cdot]$.*

*For a time grid $1 = \tau_K > \cdots > \tau_0 = 0$, if the expected local consistency error against the unconditional reverse transition for any one-step jump is uniformly bounded by $\varepsilon$:*

$$\mathbb{E}_{\boldsymbol{x}_{\tau_k}}\left[\mathbb{D}_{TV}\left(f(\boldsymbol{x}_{\tau_k}, \tau_k), \, E_{k-1|k}\left[f(\boldsymbol{x}_{\tau_{k-1}}, \tau_{k-1})\right]\right)\right] \leq \varepsilon \quad \text{for all } k \in \{1, \ldots, K\},$$

---

**Algorithm 1** Consistent Discrete Denoising Diffusion Training (CD3T)

---

**Input:** dataset $\mathcal{D}$, initial parameters $\theta_0$, weighting function $w(t, \delta)$, step-size schedule $\Delta_{1:T}$, EMA rate $\lambda$
**Output:** trained model parameters $\theta$
Initialize parameters $\theta \leftarrow \theta_0$
Initialize EMA parameters $\bar{\theta} \leftarrow \theta$
**for** each step size $\delta_i \sim \Delta_{1:T}$ **do**
    Sample timestep $t \sim p(t)$
    Set $s \leftarrow t - \delta_i$, where $s \sim p(s \mid t, \delta_i)$
    Sample data point $\boldsymbol{x}_0 \sim \mathcal{D}$
    Sample forward process state

$$\boldsymbol{x}_t \sim q(\boldsymbol{x}_t \mid \boldsymbol{x}_0) = \mathrm{Cat}(\boldsymbol{x}_t; \boldsymbol{x}_0 \boldsymbol{Q}_{1:t})$$

    Sample intermediate state

$$\boldsymbol{x}_s \sim q(\boldsymbol{x}_s \mid \boldsymbol{x}_t, \boldsymbol{x}_0) = \mathrm{Cat}\left(\boldsymbol{x}_s; \frac{(\boldsymbol{x}_0 \boldsymbol{Q}_{1:s}) \odot (\boldsymbol{x}_t \boldsymbol{Q}_{s+1:t}^\top)}{\langle \boldsymbol{x}_0 \boldsymbol{Q}_{1:t}, \boldsymbol{x}_t \rangle}\right)$$

    Compute consistency loss

$$\mathcal{L}(\theta, \bar{\theta}) = w(t, \delta_i)\, \mathbb{D}\big(f_\theta(\boldsymbol{x}_t, t) \,\big\|\, \mathrm{sg}[f_{\bar{\theta}}(\boldsymbol{x}_s, s)]\big)$$

    Update parameters

$$\theta \leftarrow \theta - \eta \nabla_\theta \mathcal{L}(\theta, \bar{\theta})$$

    Update EMA parameters

$$\bar{\theta} \leftarrow \lambda \bar{\theta} + (1 - \lambda)\theta$$

**end for**
**return** $\theta$

---

*then the global expected error between the boundary points $\tau_m$ and $\tau_K$ on the grid is linearly bounded by:*

$$\mathbb{E}_{\boldsymbol{x}_{\tau_K}} \left[ \mathbb{D}_{TV}\big(f(\boldsymbol{x}_{\tau_K}, \tau_K),\ E_{m|K}\big[f(\boldsymbol{x}_{\tau_m}, \tau_m)\big]\big) \right] \leq (K - m)\varepsilon.$$

*Proof.* The proof proceeds by a recursive application of the triangle inequality, leveraging the convexity of norms and the law of total expectation. For clarity, let $f_k \equiv f(\boldsymbol{x}_{\tau_k}, \tau_k)$. Because discrete diffusion forms a true Markov chain, the unconditional reverse transitions inherently satisfy the Chapman-Kolmogorov equation. Therefore, the expectation of the target can be written as a telescoping sequence of conditional expectations:

$$E_{m|K}[f_m] = E_{K-1|K} \circ E_{K-2|K-1} \circ \cdots \circ E_{m|m+1}[f_m].$$

Let $\mathcal{E}(k, m) = \mathbb{E}_{\boldsymbol{x}_{\tau_k}} \left[ \mathbb{D}_{TV}\big(f_k, E_{m|k}[f_m]\big) \right]$ be the expected global error from step $k$ to $m$. We establish a recursive bound. Consider the error at step $k$:

$$\begin{aligned}
\mathbb{D}_{TV}\big(f_k, E_{m|k}[f_m]\big) &= \mathbb{D}_{TV}\big(f_k, E_{k-1|k}[E_{m|k-1}[f_m]]\big) \\
&\leq \mathbb{D}_{TV}\big(f_k, E_{k-1|k}[f_{k-1}]\big) + \mathbb{D}_{TV}\big(E_{k-1|k}[f_{k-1}], E_{k-1|k}[E_{m|k-1}[f_m]]\big) &\text{(Triangle Ineq.)} \\
&\leq \mathbb{D}_{TV}\big(f_k, E_{k-1|k}[f_{k-1}]\big) + E_{k-1|k}\big[\mathbb{D}_{TV}\big(f_{k-1}, E_{m|k-1}[f_m]\big)\big] &\text{(Jensen's Ineq.)}
\end{aligned}$$

The second step mathematically relies on the joint convexity of the Total Variation distance. By Jensen's inequality, we can pull the expectation $E_{k-1|k}$ outside: $\mathbb{D}_{TV}(A, \mathbb{E}[B]) \leq \mathbb{E}[\mathbb{D}_{TV}(A, B)]$.

Now, taking the outer expectation $\mathbb{E}_{\boldsymbol{x}_{\tau_k}}$ over the entire inequality, and applying the law of total expectation $\mathbb{E}_{\boldsymbol{x}_{\tau_k}}[E_{k-1|k}[\cdot]] = \mathbb{E}_{\boldsymbol{x}_{\tau_{k-1}}}[\cdot]$, we obtain:

$$\begin{aligned}
\mathcal{E}(k, m) &\leq \mathbb{E}_{\boldsymbol{x}_{\tau_k}} \left[ \mathbb{D}_{TV}\big(f_k, E_{k-1|k}[f_{k-1}]\big) \right] + \mathbb{E}_{\boldsymbol{x}_{\tau_{k-1}}} \left[ \mathbb{D}_{TV}\big(f_{k-1}, E_{m|k-1}[f_m]\big) \right] \\
&= \varepsilon + \mathcal{E}(k - 1, m)
\end{aligned}$$

---

**Algorithm 2** Masked Consistent Diffusion Language Model (M-CDLM)

---

**Input:** dataset $\mathcal{D}$, initial parameters $\theta_0$, step-size schedule $\Delta_{1:T}$, EMA rate $\lambda$
**Output:** trained model parameters $\theta$
Initialize parameters $\theta \leftarrow \theta_0$
Initialize EMA parameters $\tilde{\theta} \leftarrow \theta$
**for** each step size $\delta_t \in \Delta_{1:T}$ **do**
  Sample sequence
$$\boldsymbol{x}_0 = (\boldsymbol{x}_0^1, \dots, \boldsymbol{x}_0^L) \sim \mathcal{D}, \quad \boldsymbol{x}_0^i \in \mathcal{V}$$

  Sample corrupted sequence $\boldsymbol{x}_t \sim q(\boldsymbol{x}_t \mid \boldsymbol{x}_0)$, where
$$q(\boldsymbol{x}_t^i = \boldsymbol{k} \mid \boldsymbol{x}_0^i) = \begin{cases} 1 - t & \text{if } \boldsymbol{k} = \boldsymbol{x}_0^i, \\ t & \text{if } \boldsymbol{k} = \texttt{[MASK]}, \\ 0 & \text{otherwise.} \end{cases}$$

  Sample intermediate state $\boldsymbol{x}_s \sim q(\boldsymbol{x}_s \mid \boldsymbol{x}_t, \boldsymbol{x}_0)$, where
$$q(\boldsymbol{x}_s^i = \boldsymbol{k} \mid \boldsymbol{x}_t^i, \boldsymbol{x}_0^i) = \begin{cases} 1 & \text{if } \boldsymbol{x}_t^i \neq \texttt{[MASK]} \text{ and } \boldsymbol{k} = \boldsymbol{x}_t^i, \\ \frac{t-s}{t} & \text{if } \boldsymbol{x}_t^i = \texttt{[MASK]} \text{ and } \boldsymbol{k} = \boldsymbol{x}_0^i, \\ \frac{s}{t} & \text{if } \boldsymbol{x}_t^i = \texttt{[MASK]} \text{ and } \boldsymbol{k} = \texttt{[MASK]}, \\ 0 & \text{otherwise.} \end{cases}$$

  Compute consistency loss
$$\mathcal{L}(\theta) = \frac{1}{\delta_t} \sum_{i: \boldsymbol{x}_t^i = \texttt{[MASK]}} \mathbb{D}_{\text{JSD}}\big(f_\theta(\boldsymbol{x}_t, t)_i \,\big\|\, \text{sg}\big[f_{\tilde{\theta}}(\boldsymbol{x}_s, s)_i\big]\big)$$

  Update parameters
$$\theta \leftarrow \theta - \eta \nabla_\theta \mathcal{L}(\theta)$$

  Update EMA parameters
$$\tilde{\theta} \leftarrow \lambda \tilde{\theta} + (1 - \lambda)\theta$$

**end for**
**return** $\theta$

---

By unrolling this recursive relationship $\mathcal{E}(k, m) \leq \varepsilon + \mathcal{E}(k - 1, m)$ from $k = K$ down to $m + 1$, and noting that $\mathcal{E}(m, m) = \mathbb{E}[\mathbb{D}_{\text{TV}}(f_m, f_m)] = 0$, the final bound evaluates strictly to:

$$\mathcal{E}(K, m) \leq (K - m)\varepsilon.$$

Because both the predictor and the expectation target depend exclusively on the observable state $\boldsymbol{x}_{\tau_k}$, an expressive neural network can theoretically drive the tracking error $\varepsilon \to 0$, rendering the bound mathematically realizable. $\qquad \square$

*Remark* C.2 (Bounding Tracking Error via Excess Risk). While Lemma C.1 bounds global error based on the local TV tracking error $\varepsilon$, our practical training objective minimizes a statistical divergence. Because the exact discrete bridge is inherently stochastic, comparing a deterministic predictor $Q = f_k(\boldsymbol{x}_{\tau_k})$ against a stochastic target distribution $P = f_{k-1}(\boldsymbol{x}_{\tau_{k-1}})$ yields an absolute expected training loss $\mathcal{L}_{local}$ that is strictly lower-bounded by an irreducible path variance (Bayes Risk, $\mathcal{L}^* > 0$). Bounding $\varepsilon$ directly using absolute loss would thus result in a mathematically vacuous bound.

However, we can rigorously bound the true tracking error $\varepsilon$ using the concept of *excess risk*. Consider the idealized setting where training minimizes the Forward KL divergence (which uniquely satisfies Proposition 3.4). Given $\boldsymbol{x}_{\tau_k}$, the expected

loss $\mathcal{L}(Q) = \mathbb{E}_{P \sim p(\cdot | \boldsymbol{x}_{\tau_k})}[\mathrm{KL}(P \parallel Q)]$ decomposes exactly as:

$$\mathcal{L}(Q) = \mathbb{E}_{P \sim p(\cdot | \boldsymbol{x}_{\tau_k})}\big[\mathrm{KL}(P \parallel \mathbb{E}[P \mid \boldsymbol{x}_{\tau_k}])\big] + \mathrm{KL}(\mathbb{E}[P \mid \boldsymbol{x}_{\tau_k}] \parallel Q)$$
$$= \mathcal{L}^*(\boldsymbol{x}_{\tau_k}) + \mathcal{E}_{\mathrm{opt}}(Q),$$

where $\mathcal{L}^*$ is the irreducible path variance of the stochastic bridge, and $\mathcal{E}_{\mathrm{opt}}(Q)$ is the *excess optimization risk* (the error of the neural network matching the exact true mean).

By Pinsker's inequality, the true local tracking error $\varepsilon$ under TV distance is strictly bounded by the excess risk, not the absolute loss. Taking the expectation over all $\boldsymbol{x}_{\tau_k}$ and applying Jensen's inequality (concavity of $\sqrt{\cdot}$):

$$\varepsilon = \mathbb{E}_{\boldsymbol{x}_{\tau_k}}\big[\mathbb{D}_{\mathrm{TV}}(\mathbb{E}[P \mid \boldsymbol{x}_{\tau_k}], Q)\big] \leq \mathbb{E}_{\boldsymbol{x}_{\tau_k}}\left[\sqrt{\frac{1}{2}\mathrm{KL}(\mathbb{E}[P \mid \boldsymbol{x}_{\tau_k}] \parallel Q)}\right] \leq \sqrt{\frac{1}{2}\mathbb{E}_{\boldsymbol{x}_{\tau_k}}[\mathcal{E}_{\mathrm{opt}}(Q)]}.$$

Applying Lemma C.1, the global expected TV error over a $K$-step trajectory is strictly bounded by $\mathcal{E}_{\mathrm{TV}}(K, 0) \leq K\sqrt{\frac{1}{2}\mathbb{E}[\mathcal{E}_{\mathrm{opt}}]}$. This cleanly isolates the reducible optimization error from the irreducible path variance, proving that our global tracking guarantee depends purely on network capacity and optimization success. While our practical algorithm utilizes JSD as a variance-bounded structural regularizer (as discussed in Section 3.2), this KL-based derivation formally validates that the stochastic MPDC objective mathematically supports meaningful global tracking bounds.

## D. Rigorous Unification of Discrete Generative Models

In Section 3.3, we conceptually positioned CDLM as a unifying framework for discrete generative modeling. Here, we provide rigorous derivations establishing how various existing paradigms emerge as specific instantiations, analytic limits, or empirical approximations of the general CDLM framework.

To facilitate a unified analysis, we define the *Generalized Consistency Objective* over a joint coupling distribution $\Pi(\boldsymbol{x}_s, \boldsymbol{x}_t \mid \boldsymbol{x}_0)$:

$$\mathcal{L}_\Pi(\theta; t, s) = \mathbb{E}_{\boldsymbol{x}_0 \sim \mathcal{D}} \mathbb{E}_{(\boldsymbol{x}_s, \boldsymbol{x}_t) \sim \Pi(\cdot, \cdot | \boldsymbol{x}_0)} \left[ \sum_{i \in \mathcal{M}(\boldsymbol{x}_t)} w(t, \delta) \cdot \mathbb{D}\big(f_{\tilde{\theta}}(\boldsymbol{x}_s, s)_i \, \big\| \, f_\theta(\boldsymbol{x}_t, t)_i\big) \right], \tag{8}$$

where $\delta = t - s$. In CDLM, the coupling is the exact Markovian forward-backward transition: $\Pi_{\mathrm{CDLM}}(\boldsymbol{x}_s, \boldsymbol{x}_t \mid \boldsymbol{x}_0) = q(\boldsymbol{x}_t \mid \boldsymbol{x}_0)q(\boldsymbol{x}_s \mid \boldsymbol{x}_t, \boldsymbol{x}_0)$.

### D.1. Exact Equivalence to Masked Diffusion Models (MDLM)

**Proposition D.1** (MDLM as Max-Step CDLM). *Assume the CDLM objective is configured with the maximum step size $\delta = t$ (implying $s = 0$), the divergence measure $\mathbb{D}(Q\|P) := \mathrm{KL}(P\|Q)$ (Forward KL, where $P$ is the target), and the continuous-time weighting function $w(t, \delta) = -\frac{\alpha_t'}{1 - \alpha_t}$. Then, the CDLM loss is mathematically equivalent to the continuous-time Negative Evidence Lower Bound (NELBO) optimized by MDLM.*

*Proof.* Substituting $\delta = t \implies s = 0$ into the exact posterior bridge $q(\boldsymbol{x}_s \mid \boldsymbol{x}_t, \boldsymbol{x}_0)$ yields a deterministic identity mapping to the clean data:

$$q(\boldsymbol{x}_s \mid \boldsymbol{x}_t, \boldsymbol{x}_0)\big|_{s=0} = \mathbb{I}(\boldsymbol{x}_s = \boldsymbol{x}_0). \tag{9}$$

Because the intermediate state is deterministically $\boldsymbol{x}_0$, the target network evaluates at the boundary. By the structural boundary condition established in Definition 3.3, $f_{\tilde{\theta}}(\boldsymbol{x}_0, 0) = \boldsymbol{x}_0$, where $\boldsymbol{x}_0$ is the one-hot representation of the data tokens.

The expectation over $\boldsymbol{x}_s$ vanishes, and the CDLM loss simplifies to:

$$\mathcal{L}_{\mathrm{CDLM}}(\theta; t, 0) = \mathbb{E}_{\boldsymbol{x}_0, \boldsymbol{x}_t} \left[ \sum_{i \in \mathcal{M}(\boldsymbol{x}_t)} -\frac{\alpha_t'}{1 - \alpha_t} \cdot \mathrm{KL}\big(\boldsymbol{x}_0^i \, \big\| \, f_\theta(\boldsymbol{x}_t, t)_i\big) \right]. \tag{10}$$

Because the true data distribution $\boldsymbol{x}_0^i$ is a deterministic one-hot vector, its entropy is zero ($\text{H}(\boldsymbol{x}_0^i) = 0$). The Forward KL divergence exactly reduces to the Cross-Entropy (CE) loss:

$$\text{KL}\big(\boldsymbol{x}_0^i \,\big\|\, f_\theta(\boldsymbol{x}_t, t)_i\big) = \text{CE}\big(\boldsymbol{x}_0^i, f_\theta(\boldsymbol{x}_t, t)_i\big) = -\log f_\theta(\boldsymbol{x}_t, t)_{i, \boldsymbol{x}_0^i} = -\log\langle f_\theta(\boldsymbol{x}_t, t)_i, \boldsymbol{x}_0^i\rangle. \tag{11}$$

Substituting this yields:

$$\mathcal{L}_{\text{CDLM}}(\theta; t, 0) = \mathbb{E}_{\boldsymbol{x}_0, \boldsymbol{x}_t}\left[ -\frac{\alpha_t'}{1 - \alpha_t} \sum_{i \in \mathcal{M}(\boldsymbol{x}_t)} \big( -\log\langle f_\theta(\boldsymbol{x}_t, t)_i, \boldsymbol{x}_0^i\rangle\big)\right] = \mathbb{E}_{\boldsymbol{x}_0, \boldsymbol{x}_t}\left[ \frac{\alpha_t'}{1 - \alpha_t} \sum_{i \in \mathcal{M}(\boldsymbol{x}_t)} \log\langle f_\theta(\boldsymbol{x}_t, t)_i, \boldsymbol{x}_0^i\rangle\right], \tag{12}$$

which is identical to the continuous NELBO derived in Eq. 11 of Sahoo et al. (2024). Since $\alpha_t$ is monotonically decreasing, $\alpha_t' < 0$, so the coefficient $-\alpha_t'/(1 - \alpha_t)$ is positive and the integrand $-\frac{\alpha_t'}{1-\alpha_t}\log\langle f_\theta(\boldsymbol{x}_t, t)_i, \boldsymbol{x}_0^i\rangle$ is nonnegative. The max-step CDLM loss therefore reduces exactly to the standard time-weighted cross-entropy/NELBO objective used by MDLM, up to the sign convention used for writing the variational bound. $\qquad\square$

## D.2. Continuous Consistency Training via Stochastic Coupling

**Proposition D.2** (Stochastic Analogue of Consistency Models). *In the local-step limit $\delta \to 0$, CDLM recovers the local consistency training objective of continuous Consistency Models, substituting the deterministic Probability Flow ODE step with the exact stochastic posterior jump.*

*Proof.* In continuous domains, Consistency Training (CT) enforces local self-consistency across adjacent steps by relying on a one-step ODE solver $\Phi(\boldsymbol{x}_t, t, s)$ to approximate the Probability Flow ODE. Their consistency distillation loss (Song et al., 2023, Eq. 7) enforces matching over a deterministic coupling:

$$\Pi_{\text{CT}}(\boldsymbol{x}_s \mid \boldsymbol{x}_t) = \delta_{\text{Dirac}}\big(\boldsymbol{x}_s - \Phi(\boldsymbol{x}_t, t, s)\big). \tag{13}$$

In categorical discrete spaces, the Probability Flow ODE $\Phi$ does not exist. The true reverse transition $p(\boldsymbol{x}_s \mid \boldsymbol{x}_t)$ is intrinsically stochastic. CDLM structurally mirrors the consistency update by replacing the deterministic ODE coupling with the exact stochastic posterior bridge:

$$\Pi_{\text{CDLM}}(\boldsymbol{x}_s \mid \boldsymbol{x}_t) = \mathbb{E}_{\boldsymbol{x}_0 \sim p(\boldsymbol{x}_0 \mid \boldsymbol{x}_t)}\big[q(\boldsymbol{x}_s \mid \boldsymbol{x}_t, \boldsymbol{x}_0)\big]. \tag{14}$$

Therefore, taking the local-step limit $\delta \to 0$, CDLM acts as the mathematically rigorous generalization of Consistency Models to discrete jump processes. $\qquad\square$

## D.3. Analytical Bypass of Progressive Distillation and Shortcut Models

**Proposition D.3** (Analytic Composition of Multi-Step Bridges). *Progressive Distillation (Salimans & Ho, 2022) and Shortcut Models (Frans et al., 2025) enforce step-size consistency via empirical multi-step rollouts or recursive bootstrapping. CDLM analytically computes this multi-step composition in closed form via the Chapman-Kolmogorov semigroup property, bypassing algorithmic rollouts.*

*Proof.* Both Progressive Distillation (PD) and Shortcut Models (SM) operate on the principle that a single step of size $2\delta$ should equal two sequential steps of size $\delta$. In PD, a student model is trained to match a target derived algorithmically from two deterministic DDIM steps of a teacher model (Salimans & Ho, 2022, Algorithm 2). In SM, a step-conditioned model $s_\theta$ minimizes a self-consistency loss (Frans et al., 2025, Eq. 5) between one large step and two chained small steps:

$$\mathcal{L}_{\text{SM}} = \mathbb{E}\left[\left\|s_\theta(\boldsymbol{x}_t, t, 2\delta) - \frac{1}{2}\big(s_{\tilde{\theta}}(\boldsymbol{x}_t, t, \delta) + s_{\tilde{\theta}}(\boldsymbol{x}_{t+\delta}', t + \delta, \delta)\big)\right\|^2\right], \tag{15}$$

where $\boldsymbol{x}_{t+\delta}' = \boldsymbol{x}_t + s_{\tilde{\theta}}(\boldsymbol{x}_t, t, \delta)\delta$ uses an explicit intermediate Euler integration step.

Because continuous state spaces lack a closed-form exact marginalization for finite step sizes, these methods must explicitly instantiate and sum over the intermediate states algorithmically. However, in discrete space, true diffusion transition

matrices inherently obey the Chapman-Kolmogorov equation. The exact posterior bridge analytically marginalizes over the intermediate state $\boldsymbol{x}_{t-\delta}$:

$$q(\boldsymbol{x}_{t-2\delta} \mid \boldsymbol{x}_t, \boldsymbol{x}_0) = \sum_{\boldsymbol{x}_{t-\delta}} q(\boldsymbol{x}_{t-2\delta} \mid \boldsymbol{x}_{t-\delta}, \boldsymbol{x}_0)\, q(\boldsymbol{x}_{t-\delta} \mid \boldsymbol{x}_t, \boldsymbol{x}_0). \tag{16}$$

Let the consistency operator defined in Eq. 3 be $\mathcal{C}_{s\leftarrow t}$.

$$
\begin{aligned}
[\mathcal{C}_{t-2\delta\leftarrow t} f_{\bar\theta}](x_t) &= \mathbb{E}_{x_{t-2\delta}\sim p(\cdot\mid x_t)}[f_{\bar\theta}(x_{t-2\delta})] \\
&= \sum_{x_{t-2\delta}} \left( \sum_{x_{t-\delta}} p(x_{t-2\delta}\mid x_{t-\delta})p(x_{t-\delta}\mid x_t) \right) f_{\bar\theta}(x_{t-2\delta}) \quad \text{(by Chapman-Kolmogorov)} \\
&= \sum_{x_{t-\delta}} p(x_{t-\delta}\mid x_t) \left[ \sum_{x_{t-2\delta}} p(x_{t-2\delta}\mid x_{t-\delta})f_{\bar\theta}(x_{t-2\delta}) \right] \\
&= \mathbb{E}_{x_{t-\delta}\sim p(\cdot\mid x_t)} [[\mathcal{C}_{t-2\delta\leftarrow t-\delta} f_{\bar\theta}](x_{t-\delta})] \\
&= [\mathcal{C}_{t-\delta\leftarrow t} \circ \mathcal{C}_{t-2\delta\leftarrow t-\delta}] f_{\bar\theta}(x_t)
\end{aligned}
$$

This strict equality demonstrates that evaluating the CDLM objective over a step size $2\delta$ mathematically evaluates the exact, marginalized two-step transition. CDLM therefore intrinsically enforces the multi-step alignment constraint of PD and SM directly in a single stage, bypassing the compounding errors of algorithmic teacher rollouts and the instability of self-consistency bootstrapping. $\square$

### D.4. Two-Stage Discrete Distillation as Approximate Couplings

Recent two-stage acceleration techniques for discrete diffusion fundamentally attempt to solve the same objective as CDLM: matching predictions between $\boldsymbol{x}_t$ and an intermediate state $\boldsymbol{x}_s$. We demonstrate that these methods can be rigorously formalized as optimizing the Generalized Consistency Objective (Eq. 8) over *approximate empirical joint couplings*, whereas CDLM utilizes the *exact analytic joint coupling*.

**Proposition D.4** (SDTT as Empirical Teacher Coupling). *Self-Distillation Through Time is equivalent to a CDLM objective where the exact data conditioning $\boldsymbol{x}_0$ is replaced by an empirical teacher approximation $\hat{\boldsymbol{x}}_0$.*

*Proof.* CDLM conditions on the true data $\boldsymbol{x}_0 \sim \mathcal{D}$ to utilize the exact Markovian coupling:

$$\Pi_{\text{CDLM}}(\boldsymbol{x}_s, \boldsymbol{x}_t) = \int q(\boldsymbol{x}_t \mid \boldsymbol{x}_0)\, p_{\text{data}}(\boldsymbol{x}_0)\, q(\boldsymbol{x}_s \mid \boldsymbol{x}_t, \boldsymbol{x}_0)\, d\boldsymbol{x}_0. \tag{17}$$

As defined in Algorithm 1 of Deschenaux & Gulcehre (2025), SDTT replaces the exact marginalization over $\boldsymbol{x}_0$ with an ancestral rollout from a pre-trained teacher model $p_{\text{teacher}}$. The teacher predicts an empirical clean state $\hat{\boldsymbol{x}}_0$, from which $\boldsymbol{x}_s$ is subsequently sampled. The effective coupling becomes:

$$\Pi_{\text{SDTT}}(\boldsymbol{x}_s, \boldsymbol{x}_t) = \int q(\boldsymbol{x}_t \mid \boldsymbol{x}_0)\, p_{\text{data}}(\boldsymbol{x}_0) \left( \sum_{\hat{\boldsymbol{x}}_0} p_{\text{teacher}}(\hat{\boldsymbol{x}}_0 \mid \boldsymbol{x}_t)\, q(\boldsymbol{x}_s \mid \boldsymbol{x}_t, \hat{\boldsymbol{x}}_0) \right) d\boldsymbol{x}_0. \tag{18}$$

Comparing $\Pi_{\text{CDLM}}$ and $\Pi_{\text{SDTT}}$, SDTT perfectly matches CDLM *if and only if* $p_{\text{teacher}}(\hat{\boldsymbol{x}}_0 \mid \boldsymbol{x}_t)$ is a perfect oracle for the true posterior $p_{\text{data}}(\boldsymbol{x}_0 \mid \boldsymbol{x}_t)$. By utilizing the actual $\boldsymbol{x}_0$ directly from the dataset during single-stage training, CDLM naturally acts as the oracle-limit of SDTT, eliminating the compounding approximation errors of the teacher network. $\square$

**Proposition D.5** (DUO-DCD as Comonotonic Latent Coupling). *Discrete Consistency Distillation (DCD) in DUO constructs a deterministic, comonotonic pseudo-bridge that violates the conditional independence of true discrete Markov transitions.*

*Proof.* As defined in Eq. 18 of Sahoo et al. (2025), DUO-DCD maps continuous ODE states to the discrete domain via Deterministic Discrete Trajectories ($\mathcal{P}_{DDT}$). It shares a single continuous latent Gaussian noise vector $\epsilon \sim \mathcal{N}(\mathbf{0}, \boldsymbol{I}_K)$ across timesteps:

$$\boldsymbol{x}_t^l = \arg\max \left( \bar\alpha_t \boldsymbol{x}_0^l + \sqrt{1 - \bar\alpha_t^2}\, \epsilon^l \right), \quad \boldsymbol{x}_s^l = \arg\max \left( \bar\alpha_s \boldsymbol{x}_0^l + \sqrt{1 - \bar\alpha_s^2}\, \epsilon^l \right). \tag{19}$$

By sharing the exact same continuous noise vector $\epsilon$ for both $t$ and $s$, DCD defines an implicit joint coupling distribution:

$$\Pi_{\text{DCD}}(\boldsymbol{x}_s, \boldsymbol{x}_t \mid \boldsymbol{x}_0) = \int \mathcal{N}(\epsilon; \boldsymbol{0}, \boldsymbol{I}_K) \, \mathbb{I}\Big(\boldsymbol{x}_t = \arg\max_t(\boldsymbol{x}_0, \epsilon)\Big) \, \mathbb{I}\Big(\boldsymbol{x}_s = \arg\max_s(\boldsymbol{x}_0, \epsilon)\Big) d\epsilon. \tag{20}$$

This forms a deterministic, *comonotonic coupling*: given $\boldsymbol{x}_0$ and the shared latent $\epsilon$, the transition between $\boldsymbol{x}_t$ and $\boldsymbol{x}_s$ is perfectly correlated by the continuous space geometry. Conversely, the true discrete forward process strictly requires that categorical transitions be conditionally independent Markov jumps. CDLM respects this structural integrity by enforcing consistency exclusively along the true discrete Markovian paths $q(\boldsymbol{x}_s \mid \boldsymbol{x}_t, \boldsymbol{x}_0)$. This mathematical distinction rigorously explains why CDLM avoids the mode-collapse and diversity loss (low generation entropy) empirically observed in DUO-DCD's deterministic projection. $\qquad\square$

# E. Additional Proofs

### E.1. Proof of Lemma 3.1

We seek to derive the probability vector for the categorical distribution $q(\boldsymbol{x}_s \mid \boldsymbol{x}_t, \boldsymbol{x}_0)$. From the definition of conditional probability, we have:

$$q(\boldsymbol{x}_s \mid \boldsymbol{x}_t, \boldsymbol{x}_0) = \frac{q(\boldsymbol{x}_t \mid \boldsymbol{x}_s, \boldsymbol{x}_0) \, q(\boldsymbol{x}_s \mid \boldsymbol{x}_0)}{q(\boldsymbol{x}_t \mid \boldsymbol{x}_0)}$$

The forward process is a Markov chain, meaning the state at time $t$ depends only on the state at time $s$ (for $s < t$), not on earlier states like $\boldsymbol{x}_0$. Therefore, the likelihood term simplifies:

$$q(\boldsymbol{x}_t \mid \boldsymbol{x}_s, \boldsymbol{x}_0) = q(\boldsymbol{x}_t \mid \boldsymbol{x}_s)$$

This gives us the proportional relationship:

$$q(\boldsymbol{x}_s \mid \boldsymbol{x}_t, \boldsymbol{x}_0) \propto q(\boldsymbol{x}_t \mid \boldsymbol{x}_s) \, q(\boldsymbol{x}_s \mid \boldsymbol{x}_0)$$

**Vector Formulation.** We now express the terms on the right-hand side using their categorical probability vectors. Let $\boldsymbol{p}(\cdot)$ denote the probability vector of a distribution.

- The prior probability of $\boldsymbol{x}_s$ is given by the forward marginal: $\boldsymbol{p}(\boldsymbol{x}_s \mid \boldsymbol{x}_0) = \boldsymbol{x}_0 \boldsymbol{Q}_{1:s}$.

- The likelihood of $\boldsymbol{x}_t$ given $\boldsymbol{x}_s$ is determined by the transitions from $s$ to $t$. The probability vector is $\boldsymbol{p}(\boldsymbol{x}_t \mid \boldsymbol{x}_s) = \boldsymbol{x}_s \boldsymbol{Q}_{s+1:t}$.

The expression $q(\boldsymbol{x}_t \mid \boldsymbol{x}_s) \, q(\boldsymbol{x}_s \mid \boldsymbol{x}_0)$ gives the joint probability $q(\boldsymbol{x}_t, \boldsymbol{x}_s \mid \boldsymbol{x}_0)$. To find the probability vector for $q(\boldsymbol{x}_s \mid \boldsymbol{x}_t, \boldsymbol{x}_0)$, we consider the probability of a specific one-hot vector outcome for $\boldsymbol{x}_s$. This is proportional to the probability of that outcome under the prior, multiplied by the probability of observing $\boldsymbol{x}_t$ given that outcome. In vector form, this product corresponds to an element-wise (Hadamard) product of the prior probability vector and the likelihood vector.

The likelihood vector, representing $p(\boldsymbol{x}_t \mid \boldsymbol{x}_s = v_i)$ for all possible states $v_i$, is given by $\boldsymbol{x}_t \boldsymbol{Q}_{s+1:t}^{\top}$. Thus, the unnormalized probability vector for $\boldsymbol{x}_s$ is:

$$\boldsymbol{p}_{\text{unnormalized}}(\boldsymbol{x}_s \mid \boldsymbol{x}_t, \boldsymbol{x}_0) = (\boldsymbol{x}_0 \boldsymbol{Q}_{1:s}) \odot (\boldsymbol{x}_t \boldsymbol{Q}_{s+1:t}^{\top})$$

The normalizing constant is the marginal probability of the evidence, $q(\boldsymbol{x}_t \mid \boldsymbol{x}_0)$. For the specific observed outcome $\boldsymbol{x}_t$, this probability is $\langle \boldsymbol{x}_0 \boldsymbol{Q}_{1:t}, \boldsymbol{x}_t \rangle$. Dividing the unnormalized vector by this scalar gives the final probability vector, completing the proof for the main formula.

**Semigroup Property.** The transitive composition follows from the law of total probability and the Markov property:

$$q(\boldsymbol{x}_u \mid \boldsymbol{x}_t, \boldsymbol{x}_0) = \sum_{\boldsymbol{x}_s} q(\boldsymbol{x}_u, \boldsymbol{x}_s \mid \boldsymbol{x}_t, \boldsymbol{x}_0)$$

$$= \sum_{\boldsymbol{x}_s} q(\boldsymbol{x}_u \mid \boldsymbol{x}_s, \boldsymbol{x}_t, \boldsymbol{x}_0) \, q(\boldsymbol{x}_s \mid \boldsymbol{x}_t, \boldsymbol{x}_0)$$

$$= \sum_{\boldsymbol{x}_s} q(\boldsymbol{x}_u \mid \boldsymbol{x}_s, \boldsymbol{x}_0) \, q(\boldsymbol{x}_s \mid \boldsymbol{x}_t, \boldsymbol{x}_0)$$

Substituting the bridge formula (Eq. 2) for each term and simplifying demonstrates that the composition holds, relying on the associativity of the transition matrices ($\boldsymbol{Q}_{u+1:s}\boldsymbol{Q}_{s+1:t} = \boldsymbol{Q}_{u+1:t}$). $\qquad\square$

### E.2. Proof of Proposition 3.4

We work within the factorized predictor class and prove fixed-point and uniqueness coordinatewise; the joint statement then follows by independence across positions. We prove both the fixed-point property and optimality.

**Fixed point.** Fix any timestep pair $0 < s < t \leq 1$, any observed noisy sequence $\boldsymbol{x}_t$, and any token position $i$. Recall that $f^*(\boldsymbol{x}_t, t)_i$ is the per-position posterior marginal, i.e., for any token value $v \in \mathcal{V}$,

$$f^*(\boldsymbol{x}_t, t)_i(v) = p(x_0^i = v \mid \boldsymbol{x}_t).$$

Let $\boldsymbol{x}_0$ be drawn from the joint posterior $p(\boldsymbol{x}_0 \mid \boldsymbol{x}_t)$ and then draw an intermediate state $\boldsymbol{x}_s$ from the analytic posterior bridge $q(\boldsymbol{x}_s \mid \boldsymbol{x}_t, \boldsymbol{x}_0)$. By construction, the resulting joint distribution equals the true conditional joint of $(\boldsymbol{x}_0, \boldsymbol{x}_s)$ given $\boldsymbol{x}_t$ under the forward process:

$$p(\boldsymbol{x}_0, \boldsymbol{x}_s \mid \boldsymbol{x}_t) = p(\boldsymbol{x}_0 \mid \boldsymbol{x}_t) \, q(\boldsymbol{x}_s \mid \boldsymbol{x}_t, \boldsymbol{x}_0).$$

Therefore, marginalizing out $\boldsymbol{x}_0$ yields the correct conditional distribution of $\boldsymbol{x}_s$ given $\boldsymbol{x}_t$:

$$p(\boldsymbol{x}_s \mid \boldsymbol{x}_t) = \mathbb{E}_{\boldsymbol{x}_0 \sim p(\boldsymbol{x}_0 \mid \boldsymbol{x}_t)}\big[q(\boldsymbol{x}_s \mid \boldsymbol{x}_t, \boldsymbol{x}_0)\big].$$

Now, for any token value $v \in \mathcal{V}$, applying the law of total probability over the intermediate state $\boldsymbol{x}_s$:

$$p(x_0^i = v \mid \boldsymbol{x}_t) = \sum_{\boldsymbol{x}_s} p(x_0^i = v, \boldsymbol{x}_s \mid \boldsymbol{x}_t) = \sum_{\boldsymbol{x}_s} p(\boldsymbol{x}_s \mid \boldsymbol{x}_t) \, p(x_0^i = v \mid \boldsymbol{x}_s)$$

$$= \mathbb{E}_{\boldsymbol{x}_s \sim p(\boldsymbol{x}_s \mid \boldsymbol{x}_t)}\big[ f^*(\boldsymbol{x}_s, s)_i(v) \big],$$

which is exactly the per-position global multi-path consistency condition. The boundary condition $f^*(\boldsymbol{x}_0, 0) = \boldsymbol{x}_0$ is trivially satisfied by the forward process formulation.

**Uniqueness under mean-eliciting divergences.** Fix $(\boldsymbol{x}_t, t)$ and a position $i$. Define the random per-position target distribution

$$P_i \equiv f^*(\boldsymbol{x}_s, s)_i = p(x_0^i \mid \boldsymbol{x}_s) \qquad \text{with} \quad \boldsymbol{x}_0 \sim p(\boldsymbol{x}_0 \mid \boldsymbol{x}_t), \ \boldsymbol{x}_s \sim q(\boldsymbol{x}_s \mid \boldsymbol{x}_t, \boldsymbol{x}_0).$$

Consider minimizing the conditional expected consistency loss at $(\boldsymbol{x}_t, t)$ over per-position predictors $Q_i \in \Delta^{|\mathcal{V}|}$:

$$\mathcal{J}(Q_i) \ := \ \mathbb{E}\big[\mathbb{D}(P_i \,\|\, Q_i)\,\big|\,\boldsymbol{x}_t, t\big].$$

For a strictly proper *mean-eliciting* scoring rule $\mathbb{D}$ (e.g., Bregman divergences such as the Forward KL divergence or cross-entropy, where $P_i$ is the target and $Q_i$ is the predictor), the Bayes act that uniquely minimizes $\mathcal{J}(Q_i)$ is exactly the expected arithmetic mean (mixture) distribution $\bar{P}_i := \mathbb{E}[P_i \mid \boldsymbol{x}_t, t]$.

For instance, when $\mathbb{D}(P_i\|Q_i)$ is the Forward KL divergence, we can analytically decompose the expected divergence as $\mathbb{E}[\mathrm{KL}(P_i\|Q_i)] = \mathbb{E}[-\mathrm{H}(P_i)] + \mathrm{CE}(\bar{P}_i, Q_i)$. Since the expected entropy of $P_i$ is independent of $Q_i$, minimizing the expected KL is structurally equivalent to minimizing the cross-entropy against the arithmetic mean $\bar{P}_i$, uniquely yielding $Q_i = \bar{P}_i$.

By the fixed-point part proved above,

$$\bar{P}_i = \mathbb{E}_{\boldsymbol{x}_s \sim p(\boldsymbol{x}_s | \boldsymbol{x}_t)} \big[ f^*(\boldsymbol{x}_s, s)_i \big] = f^*(\boldsymbol{x}_t, t)_i.$$

Hence $f^*(\boldsymbol{x}_t, t)_i$ uniquely minimizes the conditional expected loss for each $(\boldsymbol{x}_t, t, i)$. Taking expectation over $(\boldsymbol{x}_t, t)$ and summing over positions yields that $f^*$ is the unique minimizer of the overall expected consistency objective within the factorized class.

**Role of the boundary anchor.** The argument above identifies $f^*$ within the family of fixed points of $\mathcal{C}_{s \leftarrow t}$ once the cleaner-side target is anchored to the data distribution. Without such an anchor, the unanchored self-consistency loss $\mathcal{L}_{\text{cons}}(f)$ in Eq. 5 admits degenerate path-invariant fixed points: e.g., any constant-in-$t$ predictor trivially satisfies $f(\boldsymbol{x}_t, t) = [\mathcal{C}_{s \leftarrow t} f](\boldsymbol{x}_t, t)$ for $0 < s < t$, since the operator on a constant function returns that same constant. Including the boundary edge $s = 0$—equivalently, the max-step diffusion anchor $\mathbb{E}[\mathbb{D}(f(\boldsymbol{x}_t, t) \| \boldsymbol{x}_0)]$ in Eq. 7—grounds the consistency equations at the data, eliminates these degenerate solutions, and selects the Bayes posterior marginal $f^*$ as the unique population minimizer. The boundary anchor is therefore not merely a stabilizer but a population-level identification term. $\square$

# F. Additional Results

## F.1. Implementation Details

To ensure rigorous and fair benchmarking, our implementation builds upon the standard open-source frameworks established by recent state-of-the-art methods, specifically incorporating components from MDLM (Sahoo et al., 2024) and DUO (Sahoo et al., 2025). We adopt their widely used data pipelines and evaluation harnesses to guarantee that all comparisons ranging from architecture specifications to dataset processing remain controlled.

All models in this work, including baselines and our proposed CDLM variants, were trained under identical conditions to isolate the impact of the multi-path consistency objective. We utilize a standard transformer backbone with approximately 110 million parameters, consistent with the primary configurations in prior discrete diffusion literature. Training was conducted on the OpenWebText corpus (Gokaslan & Cohen, 2019) for 150,000 steps with a global batch size of 2048.

Experiments were distributed across clusters of NVIDIA A100 GPUs, with typical training runs utilizing between 64 and 256 GPUs. We employ data-parallel training with distributed gradient synchronization and automatic mixed precision to maximize throughput.

## F.2. Full Results with FP32 Sampling

Table 7 reports the generative perplexity results under 32-bit floating-point sampling, complementing the FP64 results in the main text. The trends across all models are consistent with the FP64 results.

| Model | Pretrain Steps | Distill Steps | Sampling steps with FP32 Sampling | | | | | | | | |
|---|---|---|---|---|---|---|---|---|---|---|---|
| | | | 4 | 8 | 16 | 32 | 64 | 128 | 256 | 512 | 1024 |
| *Comparison with Base Models (Trained from Scratch)* | | | | | | | | | | | |
| AR | 75K | 0 | N/A | N/A | N/A | N/A | N/A | N/A | N/A | N/A | 39.9 (5.4) |
| MDLM | 150k | 0 | 1655.2 (5.9) | 651.8 (5.9) | 255.2 (5.8) | 162.3 (5.6) | 92.1 (5.6) | 78.6 (5.4) | 57.5 (5.5) | 51.1 (5.3) | 42.4 (5.4) |
| DUO | 150k | 0 | 532.4 (5.6) | 199.6 (5.6) | 127.3 (5.7) | 96.1 (5.4) | 79.1 (5.5) | 82.4 (5.5) | 78.2 (5.4) | 73.9 (5.5) | 74.8 (5.5) |
| Ours: MCDLM | 150k | 0 | 661.1 (5.6) | 220.8 (5.4) | 118.3 (5.6) | 72.6 (5.6) | 56.3 (5.4) | 54.9 (5.4) | 35.2 (5.3) | 29.3 (5.3) | 25.7 (5.2) |
| Ours: MCDLM–PPLOptimized | 150k | 0 | 337.1 (5.2) | 117.6 (5.1) | 68.1 (5.2) | 42.4 (5.3) | 35.1 (5.2) | 24.7 (4.9) | 23.6 (5.3) | 21.2 (5.0) | 17.1 (5.2) |
| *Comparison with Distilled Models* | | | | | | | | | | | |
| MDLM + SDTT | 100k | 50k | 351.3 (5.3) | 132.5 (5.5) | 65.7 (5.2) | 44.5 (5.0) | 34.3 (5.3) | 29.9 (5.0) | 24.3 (5.0) | 21.2 (5.0) | 20.8 (4.9)* |
| DUO + DCD | 100k | 50k | 417.0 (5.4) | 172.0 (5.5) | 125.4 (5.6) | 96.2 (5.6) | 81.7 (5.5) | 85.5 (5.5) | 74.1 (5.7) | 72.3 (5.4) | 74.1 (5.3) |
| DUO + DCD (greedy) | 100k | 50k | **127.2 (4.6)*** | **111.0 (4.8)*** | 86.6 (5.0) | 72.6 (5.3) | 62.5 (5.3) | 59.8 (5.2) | 67.2 (5.4) | 61.4 (5.2) | 62.7 (5.2) |
| Ours: MCDLM + SDTT | 100k | 50k | **235.9 (5.1)** | **94.3 (5.3)** | **52.1 (5.2)** | **35.6 (5.0)** | **29.0 (5.2)** | **26.0 (4.9)** | **21.2 (5.0)** | **18.1 (5.1)** | **15.7 (4.6)*** |

*Table 7.* Perplexity (with entropy in parentheses) across different models, training setups, and FP32 sampling steps. Best results are **bolded**, second-best are underlined. * denotes entropy $< 5$, which empirically led to repetitive characters.

## F.3. MAUVE Scores for Conditional Generation

Table 8 reports the MAUVE scores for the conditional generation experiment described in Section 4.3.1.

| Model | OpenWebText | | | Lambada | | | Wikitext103 | | | PTB | | |
|---|---|---|---|---|---|---|---|---|---|---|---|---|
| | 8 Step | 64 Step | 512 Step | 8 Step | 64 Step | 512 Step | 8 Step | 64 Step | 512 Step | 8 Step | 64 Step | 512 Step |
| MDLM | 0.90 | 0.94 | 0.96 | 0.99 | 0.99 | 0.98 | 0.94 | 0.96 | 0.97 | 0.11 | 0.12 | 0.10 |
| SDTT | 0.96 | 0.96 | 0.98 | 0.99 | 0.99 | 0.99 | 0.98 | 0.99 | 0.96 | 0.07 | 0.08 | 0.07 |
| Ours: MCDLM–PPLOptimized | 0.99 | 0.97 | 0.98 | 0.99 | 0.99 | 0.99 | 0.98 | 0.99 | 0.99 | 0.02 | 0.07 | 0.07 |

*Table 8.* MAUVE $\uparrow$ scores with ancestral sampler across datasets. Since MAUVE is a distribution-based metric and 50% of unperturbed tokens serve as the condition, the models perform similarly under this saturated setting.

### F.4. Qualitative Examples

We present unconditional generation samples from four models—MDLM, DUO+DCD, SDTT, and CDLM (ours)—at three sampling budgets (4, 64, and 512 steps) to provide qualitative intuition for the quantitative metrics reported above. All examples are truncated to 1000 characters.

**Sampling steps = 4**

**MDLM**

<|endoftext|>abad in Texas Cordalo.  Gold Day\n \nThere weren very many test there of
a citizen in or recognition a Zina which directly would they need for over 49 little
of LRM, American valhouettes would thinkBut there are other sinister were trying said
5560 July Saints shipped outed, came out and something (afterne 73 LA, EEAN) was and
and really a louder cagebl hundred drove Greensboro four pissed, sacked away the same
or of the earlier Sony and large feasible, statement later made it just didn't come
out impoverished means toave.be joiningTexas, besidesasca curator book is nearly
a subscriber and daily editor within and outside LSUs.  be to making another alum
soon longtime never in the been its preferred basketball and ever was:bill speaker
of Louisianas industry, one of its most poorly conductedacted jobs in the , toosa
state of becoming.Ste blocsed across same guy on the Tuesday night naive and about
Lockntaking alerted to Ruff.  (Also Second does writing the finddesigne ...

**DUO + DCD**

"_\" would slip or top out 90 means how much these parts comes face .  what's not
even more or much far a pretty far way make the times be manufactd or far tried far
ago much much go pretty well or'and_belso the more side rather other than much in
almost Franks sake.  \" or sversely, look in Archre at least or much the more of work
much noJaAP .\" not not about it to writrtita or rather far/and the fact holds she
possibly farohott\" or or not look at \"the different kinds here far and mosteand that
somewhat much now better or more power of the User work or just much more structures,
or much its  noWrtnik .\" farll we can seem the interest of the Thomas specifically
not whom clearedch so far.  Its not or nobody't look or to handle it not he doesnt
stop formerierra  Err he looks.\"::and'and \$ or or not rather now to pay much more
much at=g-- OUT onhis basics,' print and or \"Yes,'one of''_to know out there\" not
remained a+ twisted or business lot of what Richard would ...

**SDTT**

by the same.  There is people they can do, that loos a more In-between much than the
has since the after of radio which used to be an area,\n more low than the pros among
a are basketball, baseball and the, and what of today did they try for more, but
prior to family and college the researchers knew) that the one that would really be
a concerted effort, certainly in the country is there and.  that in Their recent New
Media almost no, assessedIn Ginn,, in March, an, Ferris brothers found.  everything
from their baby toddlers, did not look good.  There was enough still for twice.  they
published a review in which mentioned the risk.  and, in the form "The Harris' his
still 'm risk was diminished most The lost lingers following the.il number were\n vers.
It, they told me the part that they were now prized was the stiffz type making Ferrier
do.  But they wrote out the awesome Adam Kruger in the factory and, when we talked to,
it the, he were in the studio and they put him many out.  The second D ...

**CDLM (Ours)**

<|endoftext|> had to live on, murder.  The hostages had one row and the two row of
the other.  (The hostage, by Kraft Foods, was 46\n \n with three armed men men in a
uniform by policeutives and certain pop artists, Racist pres Bane11 the song Delta\n
\nsterling heightened the attack aLadderem, said they had\n \n , had other hostages
thrown as they circled.  Yet the Independent\n \nnews , and the the raid's days, is
much more than when there were\n the possibilities for the same subject.  The Dec
said KERS could said, if\nthen AERS did not food New Whorig on board the rear of the
Blacks, only so on if the\n \nimmigrants were Indians and the, had most of the Blacks
Blacks not be French.  Yet the KERS then asked KERS to claim that kicking a ball for\n
\nthe comedy not in the right thing.  The Good of the Brain, the K and Bola Four of the
NRL., on the night of the Opening Cere the the Challenge Games, the Mr.  Roberto, who
players argued over farmers and pacified movement farmers the the late 70s ...

**Sampling steps = 64**

**MDLM**

<|endoftext|>, the next thing they saw\n \n was a male scream that the victim wanted
to kick something,\" Quinn said.  At that point, he got up and started screaming.\"\n
\nPolice soon pursued the cab into the bar.\n \n \"They picked up the suspect and
left.  They walked to the bathroom and they came to some dried blood coming from the
victim's mouth and they located [the suspect inside the bar],\" Quinn said.  \"They're
not taking a victim's DNA because obviously, they don't know if the male victim had
a knife or both.\"\n \nThe addition was found closed and \"several items that were
initially used in the attack, Quinn\n \nSmith said.\n \nThe man still lives near
Longleston in Sonoma Monday night and Tuesday March 23.\n \nMcNeil told KING-TV in
Southern California, where he met UT A&M student there on Saturday night after a\n
\ncar accident there.\n \nThe arrest warrant was processed Tuesday, and police hadn't
given much a negative Ephesian police report.  \"They remain positive.\"\n \n \"It's
...

**DUO + DCD**

<|endoftext|> of Posture-17.  Remember amateur domestic monitored product registration
stands at 73, and a number of matches have been canceled.\n \nCivil -rights groups
have taken part in the struggle against Amnesty and defend the Constitution, the
official Global Times reported.\n \nNortino prison executives face questions about
arrest\n \nChristian Numel, who have been charged over a July 2012 incident remains
behind bars in the Shenyang Dalian-chu jail until Liu in November, the son-in-law
of Hui, chief of several members including Prince John Roman's condense Panda
conglomerate.  Numel was acquitted on statements made during a Aug.  16 interview
by Yong Kushu, a Chinese state radio website.\n \nThe spate of border arrests has
signalled the intent of broadening of human rights in China.\n \nThe computers that
were captured looking anti-government could be easily labeled as spies by legal
experts that they were illegal, and hundreds of names of senior Chinese security
officials were ...

**SDTT**

<|endoftext|> the same thing too.  She also was inspiredShe threw together a group of
local kids.  When a lot of kids like Kyle came in to introduce herself to us, it kind
of made her realize that she genuinely wanted to go out and be involved in some way.
He was a model type of kid (that's just her imagination:  that's what Los Angeles was
for her home).  During her time in St.  Louis, she had already been in some pretty
fine little church churches.  She was one of those sort of people that I got to
surround myself with.\n Yeah, it was an extremely exciting experience.  It was enough
money for you to be behind the camera.\n After that, we did the improv concert, and I
didn't want to go and was excited.  I was nervous.  I was flying around and expecting
it to sound like it was this kind of a simulator, a real character.  And it was really
difficult as a child.  All the cameras are on the stage, and I told my director, "I
not have to screw it up.  I just have to do it." Anybody on my money, I just w ...

**CDLM (Ours)**

<|endoftext|> largest union, JPMorgan Chase-DHS Bank, decided to resign earlier this
week when the UK voted to liberalise the EU, threatening to pull the country from
the customs union.\n \nThe MEP has warned that one of the best ways to leave EU is
to leave the Greek exit, which is backed by groups such as the social-democratic
Democracy Alliance, which is a member of the Remain campaign.  He warned of further
\"political or social risk\" unions being back in the European Union, when a coalition
of government removed the backing of its politicians and staff members from its
membership.\n \nEarlier this week, the MEP had been resigning from the EU after being
alerted to questions over some of his newspaper articles.\n \nThe Dublin-born MEP told
a joint news conference in France morning that he was not following the decision to
liberalise the EU and use it to prevent the UK from abandoning membership.  This would
apparently result in a period of turbulence on the prospects of leaving ...

**Sampling steps = 512**

**MDLM**

<|endoftext|> him when [Singh was about to the] Supreme Court, and said, I hit every benches.  He took a box of 10.  Then he grabbed one and said, Okay, we're going to be OK now.\n \nToward the evening, Ahmadi, the second baseman of the 66, thrown Ram down onto a small concrete ground before spread his wings and clearing the deck for the ceremony.\n \nThe people near the lockers realised immediately that Deepak Ohera, another then policeman, in the eighth over of the sixth innings, had pushed Ram sprayed himself to the ground and fell.\n \nThey ran and chased away all those that did not know what had happened.  Dahla, who had enough power and pace to drive every one and one of the Indians home from college the next day, was killed.\n \n \"They killed me,\" he remembers.  \"It is sad.  I remember it for a minute.  Now, I feel a little more having my family and just going to work hard and play.  I have never felt that.\n \nGusmen table top scoring\n \nOnly 30-year-old 39-year-old Ali Ganes ...

**DUO + DCD**

"<|endoftext|> after a year and a million days, Sarafaz is about to do the movie, \"Jenna,\" who faces the streets of Virginia.\n \nChristine Nakabub meig:  Judith Kharfatabi did actually see the tape at first, but it was pulled from the November 4 of the initially [main show] screening, now for November 3 because it's really hard to make \"Pappers Out\" to review, like, November 14.  It's \"Noh and Tam\" -- \"It's a Link-in Land.\" And, so I consider it even tougher, four months to make.  It's dissimilar to when I talked with Lois Jordan when she was done.  I said, \"Here's exorcism.\" She said, \"Well, first, Ms.  Out, it was inside.  What does it take to do these sagas in those daysyou'll get your choice.  I'm visiting now with Fox, so it's just kind of certain I knew she's Lady Gaga.  So I'm expecting, it's not Tammy Jenna.  It's that, indeed.  He's on board, \"computer in the middle of the street,\" at 86 10th Street.  I did Apo Hammer today, on November 15, and this is a great time be ...

**SDTT**

<|endoftext|> off the floor.  The two didn't happen because we were rested.\n "My son came back and came off to the floor on the opening day out of UCLA, and while I played, I still had my skin covered up from it.  It was a long, tough opportunity to go to play by a great team, but I was kind of a little bit exposed to some of the things that we put him ready to go."\n In the end, I liked that performance against UCLA. He was able to have a very positive positive going through some of the injuries that I've had to deal with.  He went on to have an ankle and ended up leaving the game and actually went down to play.  He was very lively, and we got the win.\n "I've gotten a really good relationship with a lot of my teammates, with the leadership guys in the locker room.  With the defense guys, with other guys in the locker room, he was able to play."\n And that worked.  Check out the highlights from the season to come.\n "We haven't recovered from the line or our run or anything like that.  It was fr ...

**CDLM (Ours)**

<|endoftext|> the EU as the primary focus of civil society, and is fully respected in the EUs role in shaping the global economy.\n \nSo not only is the impact of Polish economy and investment in Poland having on the political and external context of the EU and its investment in many of the European countries, but the reality is the political and external context of the EU is a critical economic partner.  That is why the EU will continue to continue to thoroughly compete with the rest of Europe.\n \nThe EU has one of the worlds largest trading partners for one of the worlds biggest markets and trans-Atlantic European integration.  Though the EU continues to strengthen its position on the EUs global economy, it continues to overcome the challenges that Europeans face from the EU, and its financial commitment through its financial reporting and economic practices.\n \nFrance is one of the largest investments in the EU. In addition to building the international economy to be able to ...

