# OpenReview forum: "Consistent Diffusion Language Models"
_ICML.cc/2026/Conference — ICML 2026 regular_

### Official Review · Reviewer_CB2B · 2026-03-10

**Soundness:** 3
**Presentation:** 2
**Significance:** 3
**Originality:** 2
**Overall Recommendation:** 4
**Confidence:** 3

**Summary:**

This paper introduces Consistent Diffusion Language Models (CDLM), a framework designed to enable efficient, few-step generation in discrete diffusion models. Because discrete state spaces lack a probability flow ODE (PF-ODE), the authors propose Multi-Path Discrete Consistency (MPDC). Instead of mapping a deterministic trajectory, MPDC enforces prediction consistency across an ensemble of stochastic paths defined by exact posterior bridges. Empirically, CDLM demonstrates promising performance on text generation benchmarks, achieving significant speedups and outperforming strong base models (MDLM, DUO) and multi-stage distilled models (SDTT, DUO-DCD) in the few-step regime.

**Compliance With Llm Reviewing Policy:**

Affirmed.

**Final Justification:**

I increase the review to weak accept. Although it cannt have a guarantee for one-step generation, it still shows promising results.

**Key Questions For Authors:**

- How does the consistency objective in Eq. 5 work with the token-independence assumption in the one-step generation limit? Does the factored nature of $f_\theta$ place a hard upper bound on the quality of extreme few-step generation, and if so, how is this mitigated?
- Could you explicitly define the "expected consistency loss" mathematically right before or within Proposition 3.4 so the theoretical claim is self-contained before the practical objective (Eq. 5) is introduced?
- The "max-step" scheduler recovers the standard diffusion objective as a regularizer. How does the model behave if k_ms is set to zero?

**Limitations:**

The primary limitations of this paper are theoretical. While the method works excellently in the few-step regime, the methodology does not resolve the independence assumption limitation inherent to f_\theta, making the theoretical limit of one-step discrete generation via Eq. 5 questionable. Given the promising empirical results, I would like to increase my score if the theoretical limitation could be addressed.

**Strengths And Weaknesses:**

**Strongths**:

- The paper demonstrates impressive empirical performance. It achieves massive speedups (64x-128x) over standard MDLM and performs exceptionally well in the 4-to-16 step regime without requiring complex, multi-stage distillation pipelines.

**Weakness**:

- My main concern is the validity of the objective in Eq. 5, which is confusing and potentially flawed. Because the predictor $f_\theta$ inherently relies on the assumption of independence among token positions given the noisy state, the consistency loss in Eq. 5 cannot magically force the model to capture the complex joint distribution of the full sequence in a single step. This fundamental independence bottleneck for 1-step generation remains unaddressed.
- The writing can be improved:
    - The explanation following Lemma 3.1 about how bridges "compose transitively" (the semigroup property) is unclear. While checking Appendix B.1 clarifies the math, the main text lacks the necessary explanation for the reader to easily follow along.
    - In Proposition 3.4, the expected consistency loss is not formally defined prior to this proposition (the empirical loss is introduced later in Eq. 5), causing a disconnect in the narrative.

---

> ### Author Rebuttal · Authors · 2026-03-31
>
> Thank you for the careful reading and thoughtful feedback. We agree that the submitted version was somewhat loose at the point you identified. Our intended claim is narrower, and we will make it more explicit in revised manuscript: *CDLM is a few-step objective, not a claim to solve true 1-step discrete generation from maximal noise.*
>
> **1-step limit and token independence [W1/Q1/L1].**
> We agree with the core concern, even though the right scope is slightly different. Since $f_\theta(x_t,t)$ is trained through tokenwise prediction, Eq. 5 indeed does *not* by itself eliminate the well-known difficulty of exact 1-step discrete generation from maximally corrupted noise. In that extreme limit, the optimal predictor is the unconditional marginal, which is highly multimodal and a consistency loss alone cannot be expected to recover full sequence structure in a single jump. At the same time, this does *not* mean the model ignores token dependencies altogether: each token prediction conditions on the full noisy state $x_t$, and over multiple refinement steps the evolving state can progressively resolve cross-token structure. This is precisely the regime CDLM is designed for. Its role is therefore not to make a factorized predictor suddenly become an exact one-step joint generator, but to *reshape the training signal so that a finite-capacity model can make more reliable coarse denoising updates in the few-step regime* (e.g., 4–16 steps). We will revise the paper to state this distinction more explicitly and to align the theory more clearly with the empirical regime in which CDLM is intended to operate.
>
> **Writing updates for theoretical clarity [W2/Q2].**
> We will clarify this. The statement is simply Chapman–Kolmogorov consistency of bridges from the same Markov chain:
> $$q(x_u \mid x_t, x_0) = \sum_{x_s} q(x_u \mid x_s, x_0) q(x_s \mid x_t, x_0), \quad \text{for } t > s > u.$$
> That is, composing the bridges $t \to s$ and $s \to u$ and marginalizing $x_s$ recovers the direct bridge $t \to u$. We will make this explicit in the main text rather than relying on Appendix B.1.
> We further agree that Proposition 3.4 should be self-contained. The intended population objective is:
> $$\mathcal{C}\_{s \leftarrow t} f(x_t,t) := \mathbb{E}\_{x_0 \sim p(x_0 \mid x_t)} \mathbb{E}\_{x_s \sim q(x_s \mid x_t, x_0)}[f(x_s,s)],$$
> $$\mathcal{L}\_{\mathrm{cons}}(f) = \mathbb{E}\_{t>s, x_t \sim p_t} [\mathbb{D} (f(x_t,t), \mathcal{C}_{s \leftarrow t} f(x_t,t))].$$
> Eq. 5 is the empirical approximation of this objective. Under a strictly proper per-token loss (e.g., cross-entropy), the Bayes-optimal solution within the factorized class is the posterior marginal denoiser $f^*(x_t,t)=p(x_0 \mid x_t)$. We will move this definition before Proposition 3.4 so the theoretical statement is properly grounded.
>
> **Role of the max-step anchor [Q3].**
> Your intuition here is exactly correct. When $\kappa\_{\mathrm{ms}} = 0$, the model collapses: it achieves deceptively low perplexity but produces low-entropy, highly repetitive outputs. As shared in S4.4 and Table 7, the max-step term is found to be *structurally necessary*, as it anchors the consistency chain and prevents degenerate low-entropy solutions.
>
> Overall, we are quite grateful for your meticulous feedback, which we believe identified a *scope/exposition issue rather than a flaw in the few-step objective itself* and has helped strengthen the clarity of our narrative. Given your note that resolving this theoretical point would materially change your assessment, we hope these clarifications address the central issue and better reflect the validity, scope and impact of our work.

---

> > ### Author Rebuttal · Reviewer_CB2B · 2026-04-04
> >
> > Thanks for the rebuttal. Although it cannt have a guarantee for one-step generation, it still shows promising results.

---

> > > ### Author Response · Authors · 2026-04-07
> > >
> > > Thank you again for the thoughtful and technically precise feedback. We are glad the revised clarification helped address the main concern, and we appreciate your continued engagement. Your comments have been very helpful in sharpening both the scope and presentation of the work.

---

### Official Review · Reviewer_hixn · 2026-03-12

**Soundness:** 3
**Presentation:** 3
**Significance:** 3
**Originality:** 2
**Overall Recommendation:** 5
**Confidence:** 3

**Summary:**

This paper addresses few step sampling in discrete diffusion language models. Since the probability flow ODE used by continuous consistency models does not exist in discrete space, the authors enforce self-consistency across posterior bridges between noise levels. Combined with JSD divergence and a diffusion anchor loss, the method trains without a teacher. Experiments compare against base diffusion language models and two-stage distilled baselines.

**Compliance With Llm Reviewing Policy:**

Affirmed.

**Final Justification:**

The paper presents a clean and well-executed approach to few-step discrete diffusion. The ablations are thorough and the writing is clear.

My main concern was the FLOP matched comparison with MDLM. The authors ran this experiment and showed the gap persists across step counts, which is convincing evidence that the gains come from the training principle rather than additional compute. This addresses my primary weakness.

The theoretical novelty remains limited as the ingredients are known but the combination works very well. Terefore I also adjusted my score.

**Key Questions For Authors:**

1. Each training step requires two forward passes. Have you compared against MDLM matched by total FLOPs?
2. Concurrent work [1] approaches few step discrete generation by working on the continuous simplex, avoiding the missing probability flow ODE. How do the authors see the tradeoffs between the two approaches?
3. Any evidence on how the method scales beyond 110M parameters?

[1] Roos et al., "Categorical Flow Maps", 2026

**Limitations:**

yes

**Strengths And Weaknesses:**

**Strengths**
- The exact posterior bridge seems a intuitive choice for the coupling. It avoids the approximations used by SDTT (teacher rollouts) and DUO-DCD (argmax projection).
- The single stage model outperforms two-stage methods (SDTT, DUO-DCD) at most sampling budgets.
- The divergence ablation shows that JSD is necessary for teacher free training. The paper gives a clear explanation for this.
- The method promises to serve as a better base for subsequent distillation. MCDLM + SDTT outperforms the MDLM + SDTT baseline.
- The approach recovers MDLM, consistency models, and progressive distillation as special cases.

**Weaknesses**

- The main ingredients (posterior bridges, consistency training, EMA, anchor loss) are known. The combination for discrete space is well executed (especially using JSD) but the theoretical novelty seems limited.
- Each CDLM training step requires two forward passes, while MDLM requires one. The comparison is at equal training steps but not equal FLOPs.
- All experiments are at 110M scale. It would be nice to see results at a larger scale.
- Minor: typos "consistenctly" (p7), "compred" (p6), missing reference in Table 2 caption ("described as in ()")

---

> ### Author Rebuttal · Authors · 2026-03-31
>
> Thank you for the thoughtful review and for the positive overall assessment. We especially appreciate your recognition of the exact posterior bridge, the role of JSD in stable teacher-free training, and CDLM as a stronger base for distillation. Your questions help sharpen the positioning of the paper.
>
> **On novelty and theoretical contribution [W1].** We agree that several individual ingredients (posterior bridges, consistency training, EMA, anchor loss) are known. The novelty is therefore not in isolation, but in the *discrete-native synthesis*:
> 1. identifying the exact posterior bridge as the correct analogue of the missing PF-ODE object in discrete diffusion,
> 2. enabling a single-stage, teacher-free consistency objective directly on this bridge, and
> 3. unifying MDLM (as a boundary case) and discrete distillation methods under a common view.
>
> So we agree the contribution is best understood as a *new formulation and training principle for discrete few-step generation*, rather than novelty of ingredients viewed one-by-one.
>
> **FLOPs fairness and training cost [W2/Q1/L].** We agree this should be made more explicit. Each CDLM update uses an additional target-network forward pass relative to MDLM, so the per-step training cost is higher.
> To address fairness directly, we additionally trained MDLM for 2$\times$ more optimization steps, so that it consumed *strictly more total training FLOPs* than CDLM. Even under this compute-favorable budget, MDLM still underperforms CDLM in the few-step regime. We will add this comparison in the revised manuscript, as it more cleanly separates algorithmic gains from per-step compute. Also note that the intended tradeoff is that this additional compute is paid once during training, whereas the primary bottleneck we target is inference-time denoising cost. CDLM spends moderately more pretraining compute to substantially reduce generation steps.
>
> **Tradeoffs with continuous-simplex / flow-map approaches (e.g., CFM) [Q2].** We view these approaches as *complementary rather than competing*. Our reading is that methods such as CFM address the same few-step bottleneck by moving to a *continuous state space* (e.g., the simplex / continuous relaxation), which makes flow-map machinery and consistency-style self-distillation direct, and can support aggressive step reduction. CDLM instead remains *fully discrete-native*, leveraging the exact posterior bridges of the original corruption process. We therefore see the tradeoff as one of modeling choice rather than superiority. We expect both directions to coexist fruitfully—much like flow- and diffusion-based methods have coexisted in continuous generative modeling—and believe direct, compute-matched comparisons would be very valuable future work.
>
> **Evidence on scaling beyond 110M [W3/Q3].** Our choice of experimental setup is consistent with others in the literature, from established works like MDLM (NeurIPS-24), SEDD (ICML-24), BD3-LM (ICLR-25), Duo (ICML-25) to emerging literature like CARD, CANDI and DUEL (arxiv-26). While it is almost always desirable to have more empirical results, we believe we have proved our method as per the established criteria in the literature. Nonetheless, CDLM does *not* introduce a new backbone or exotic architecture: it retains the same underlying transformer-style denoiser as the base diffusion models and changes the *training objective*, not the model family. This makes us optimistic that the method should scale along with the underlying DLM family. Still, we agree that this is not a substitute for direct large-scale evidence, and we will present broader scale-up validation as an important next step rather than over-claiming it here.
>
> **Minor fixes.** We will fix the noted typos and missing reference.
>
> Overall, we appreciate your feedback as it sharpens the core message: CDLM is best viewed as a *discrete-native consistency framework* that (i) identifies the exact posterior bridge as the central object, (ii) enables effective single-stage few-step training, and (iii) provides a stronger base for subsequent distillation.

---

> > ### Author Rebuttal · Reviewer_hixn · 2026-04-04
> >
> > Thank you for the detailed response. The FLOP matched comparison with MDLM is good. At which step counts does the gap persist and where does it close? I maintain my score but the results are promising and I am open to raising it.

---

> > > ### Author Response · Authors · 2026-04-05
> > >
> > > Regarding your question on CDLM vs MDLM performance gap across step counts, we find that the **performance gap persists throughout**, even in the compute-favorable MDLM setting ($300k$ steps). Continued MDLM training (beyond $150k$ steps that we initially reported) does **not** lead to significant further improvement for the given model and data (sizes). In particular, CDLM maintains a consistent advantage in the few-step regime. The gap does narrow slightly at larger sampling budgets, which is expected since standard MDLM is naturally optimized for high-step refinement, but we do not observe a reversal on the budgets we evaluated ($1-1024$). We therefore interpret this result as evidence that the gains are not a consequence of unequal compute / training budgets, but arise from the training principle itself. We will include the full training curves in the revision to make this trend explicit.
> > >
> > > Thank you again for the thoughtful feedback and continued engagement. We also appreciate your openness to revisiting the score, and hope this additional clarification is helpful for your final assessment.

---

### Official Review · Reviewer_kQ7T · 2026-03-16

**Soundness:** 3
**Presentation:** 3
**Significance:** 3
**Originality:** 3
**Overall Recommendation:** 5
**Confidence:** 3

**Summary:**

The paper proposes a form of self-distillation for training discrete diffusion models.
- For standard masked diffusion, the training objective is to predict tokens before denoising, ie predict $x_0$ given $x_t$.
- Instead, the proposed method minimizes the difference between the an EMA teacher's prediction of $x_0$ at a less noised sample $x_s$ given a student's more noised sample $x_t$

The paper shows moderate performance improvements against DUO and significant performance improvements against Masked diffusion language models at the 110M-parameter scale.

**Compliance With Llm Reviewing Policy:**

Affirmed.

**Final Justification:**

I have increased my confidence as a result of the compelling FLOPs fairness experiments.

**Key Questions For Authors:**

Can you provide some ablation that accounts for flops expended (for example, validation perplexity across the training run, both in terms of training steps and in terms of approximate flops)?

Could you clarify: what property do you prove that your objective has that the MDLM objective does not?

**Limitations:**

The drawbacks of the method in terms of FLOPs and memory usage should be discussed.

**Strengths And Weaknesses:**

Strengths
- (Soundness) The experimental setup and theory is generally sound, aside from a couple of omissions I detail below.
- (Presentation) Aside from a few minor nitpicks (see below), the paper is well-written and was easy to understand.
- (Originality) The proposed method of using an EMA teacher on a sample with less noise on a sample with more noise is novel as applied to DLMs. A similar method has been found to be successful for vision-language embeddings (see SILC [1]).
- (Significance) If the performance improvements generalize to a large pretraining setup, they would represent a significant improvement on the state-of-the-art for diffusion language models.

Weaknesses:
- (Soundness) The baselines are _not_ flops-equivalent, the proposed method uses ~33% more compute per training step. This is important for pretraining. The proposed method requires two forward passes and one backward pass per training step (so flops = 8ND), versus standard MDLM which uses one forward pass and one backward pass (flops = 6ND). I would like to see the flops-matched MDLM vs MCDLM baseline.
- (Soundness)  The proposed method also uses 2x memory when compared to MDLM. This limitation is also not discussed in the paper, although less important for pretraining.
-( Soundness)  It is not clear to me how the proposed theory and links to probability flow ODEs explains why that the method outperforms MDLM. Doesn't minimizing the MDLM objective also give a method which satisfies Global Multi-Path consistency/

Nitpicks:
- I believe there is an additional "[" in equation (4)
- I believe there is a missing $x_s$ in the marginalization variables in equation (5)


[1] Naeem, Muhammad Ferjad, et al. "SILC: Improving vision language pretraining with self-distillation." European conference on computer vision 2024.

---

> ### Author Rebuttal · Authors · 2026-03-31
>
> Thank you for the careful review and for the positive overall assessment. We especially appreciate your questions on compute fairness and the precise relationship between CDLM and MDLM, addressing which makes the contribution clearer.
>
> **Compute / FLOPs fairness and memory cost [W1/W2/Q1/L1].**
> We agree this is an important constraint and should be stated explicitly. Each CDLM update includes an additional target-network forward pass, yielding higher per-step cost (approximately $8ND$ vs. $6ND$ token-FLOPs for MDLM). CDLM also incurs higher training-time memory due to maintaining the EMA target and its activations. To address fairness directly, we trained MDLM for 2$\times$ more optimization steps, so that it consumed *strictly more total training FLOPs* than CDLM. Even under this compute-favorable budget, MDLM still underperforms CDLM in the few-step regime. We will add this comparison in the revised manuscript.
> Also note that the intended tradeoff is that this additional compute is *paid once during training*, whereas the primary bottleneck we target is inference-time denoising cost. In deployment, CDLM spends modestly more pretraining compute to substantially reduce the number of refinement steps required at generation time.
>
> **What CDLM provides beyond MDLM [W3/Q2].** This is an excellent question that deserves thorough clarification. We do *not* claim that CDLM has a different *infinite-capacity* Bayes optimum than MDLM. In the ideal limit, both objectives are aligned with the same posterior denoising target $p(x_0 \mid x_t)$. In fact, MDLM appears as a boundary case of CDLM: setting $\delta = t$ (i.e., $s=0$) reduces the bridge target to the standard diffusion objective (S3.3). The distinction is therefore not in the asymptotic target, but in the *training constraints imposed on finite models*: MDLM supervises only the boundary mapping $x_t \rightarrow x_0$, while CDLM includes this boundary supervision but additionally enforces agreement across intermediate posterior bridges $x_t \rightarrow x_s \rightarrow x_0$ over a distribution of step sizes.
>
> Thus, CDLM explicitly trains the model to be path-consistent / schedule-robust, whereas MDLM does not impose this constraint during training. The gain is therefore an *inductive-bias improvement*: CDLM provides richer supervision that enables finite-capacity models to perform reliable large denoising jumps in the few-step regime. A useful way to phrase this is that MDLM is *compatible* with multi-path consistency at the optimum, but CDLM *actively enforces it during training*.
>
> **On the PF-ODE / consistency perspective.** We agree the current draft could better separate conceptual motivation from practical effect. The role of the PF-ODE analogy is not to claim that discrete diffusion inherits continuous guarantees directly, but to motivate the *missing consistency object in discrete space*. The exact posterior bridge plays this role. The empirical improvement of CDLM over MDLM is not due to a different optimal solution, but due to the *explicit enforcement of bridge consistency across multiple scales*, which directly regularizes few-step behavior.
>
> **Minor fixes.** We will fix the stray “\[” in Eq. 4. $x_s$ could be used in place of $\delta$ for more clarity but current version is also fine.
>
> Overall, we appreciate your feedback, which highlights two important clarifications: (i) the training-compute vs. inference-efficiency tradeoff, and (ii) the precise sense in which CDLM improves over MDLM, namely through stronger finite-model path-consistency constraints rather than a different asymptotic target. We hope these clarifications strengthen your confidence in both the evaluation and the conceptual contribution.

---

> > ### Author Rebuttal · Reviewer_kQ7T · 2026-04-04
> >
> > I find the idea that the method still outperforms MDLM even when MDLM is given a significant compute bump compelling and this has resolved my main concern with the paper.

---

> > > ### Author Response · Authors · 2026-04-07
> > >
> > > Thank you again for the careful reading and constructive feedback. Your comments improved how we present the method's inductive bias and compute tradeoffs, and we appreciate your support.

---

### Official Review · Reviewer_c4BZ · 2026-03-24

**Soundness:** 3
**Presentation:** 3
**Significance:** 3
**Originality:** 3
**Overall Recommendation:** 4
**Confidence:** 4

**Summary:**

In this work authors introduce consistent diffusion models, with the intuition that in the context of discrete diffusion models while we cannot introduce per-sample/particle/trajectory level consistency we could enforce disbtribution level consistency. To enable it they formulate the problem with multi-path consistency in expectation.

They introduce a clean formulation native to discrete diffusion and demonstrate strong few step generation quality outperforming previous distillation approaches in the context of diffusion LLMs.

**Compliance With Llm Reviewing Policy:**

Affirmed.

**Key Questions For Authors:**

1. While authors discuss diffusion duality/DUO style methods, how does proposed method compared to a modified version of DUO where consistency objective is enforced in expectation over multiple shared noise-samples i.e., averaged over different \epsilon? Want to better understand what empirical and conceptual design choices are more effective to serve broader community.

2. As authors touch up on diversity as one of motivation of choosing JS divergence, it would be informative to understand and report diversity metrics of different methods both at per-sample level and distributional level?

3. Given consistency methods modify underlying mapping in case of continuous diffusion, curious to understand what is effect of step-size schedule and total number of denoising steps at training and its implications on performance

4. Finally, though not asking for comparision would appreciate authors perspective on recent works like Flow-based Language Models (FLM, arXiv:2602.16813) which proposes continuous denoising over one-hot representations and avoids analytic discrete posteriors entirely while enabling one-step generation. As in FLMs can assume PF-ODE but also satify distributional consistency i.e., expectation over paths similar to proposed work but with weaker assumptions it would be informative to better understand tradeoffs.

**Limitations:**

Discussed in weakness

**Strengths And Weaknesses:**

### strengths

- Enforcing consistency in expectation makes sense in the context of discrete diffusion inspired by path integrals, etc.
- Clean and unified formulation of distributional consistency for discrete diffusion models, acceleration.
- Strong emperical performance especially in few step regime even outperforming distilled models
- Paper is clearly written and easy to follow

### Weakness and Limitation
- The necessity of having an analytic discrete posterior bride is limiting factor of this work but dominant forward processes like masked and uniform diffusion satisfy it but unclear on broader applicability.
- While performance of current model would depend on number of sampling steps at training time, dont see any systemic study w.r.t this hyperparameter and other desing choices.
- How sensitive are training design choices w.r.t max-step, JS divergence, EMA targets etc compared to training other dLLMs.

---

> ### Author Rebuttal · Authors · 2026-03-31
>
> Thank you for the thoughtful review and for engaging with both the conceptual framing and empirical design.
>
> **On broader applicability of assumption [W1].**
> We agree that requiring an analytic posterior bridge is a key modeling assumption. However, this already covers broad corruption families and—unlike many recent approaches tied to a single corruption family—we explicitly instantiate and illustrate benefits of CDLM on two dominant forward processes. Since these corruption families underpinning much of the discrete denoising diffusion [1] literature have been successfully applied across NLP [1], (quantized) vision [1], graphs / molecules [2], music [3], and scientific systems [4], CDLM inherits this scope rather than introducing a new restriction. In this sense, our method is aligned with—and directly applicable wherever—standard discrete diffusion models are used.
>
> **Training-time steps clarification and design choices [W2/W3/Q3].**
> This is an important point, and we realize it should be clearer. Like MDLM, CDLM is trained in continuous time ($T=\infty$ setup) and is not tied to a fixed inference budget. This is a prevalent setting found effective in prior works (e.g., Table 11 in MDLM paper). CDLM additionally samples an interval ($\delta = t-s$), so training controls a *distribution over jump sizes*, not a single step count. This is also why the scheduler ablations matter. The appendix already includes a systematic study (Table 6), and the effect is meaningful rather than erratic: changing the distribution over $\delta$ predictably shifts where the model performs best. We will surface this more clearly in the final manuscript. More broadly, our view is that $p(\delta)$ can be interpreted as a *curriculum over denoising difficulty*: shorter jumps emphasize local consistency, while larger jumps train the model to make reliable coarse updates in the few-step regime.
>
> In general, the sensitivities you highlight are not incidental and expose meaningful structure in the objective. The appendix ablations (S3.2, S4.4, AD.1) show these choices induce *predictable and interpretable tradeoffs*, rather than unstable behavior.
>
> **Relation to modified DUO [Q1].**
> This is an interesting comparison. Averaging a DUO-style objective over multiple shared $\epsilon$ samples would reduce Monte Carlo variance, but it would still match a different coupling. CDLM targets the *exact discrete posterior bridge* and allows for single-stage, teacher-free training, while DUO-style methods define an *empirical coupling* via shared noise and projection and is attractive for importing continuous consistency machinery. Thus the distinction is not just variance, but the underlying object. The broader takeaway is that *exact bridges are preferable when available*, while shared-noise constructions may serve as approximations otherwise. We will clarify this perspective, and also make the connection of CDLM to DUO variants (and other methods discussed in S3.3) more precise through equation-level comparisons in appendix.
>
> **Perspective on FLM [Q4].**
> We view FLM as concurrent, distinct and *complementary*. It addresses the few-step bottleneck by moving to a *continuous one-hot state space*, recovering flow-map machinery and enabling very aggressive step reduction (including 1-step generation). This effectively trades analytic discrete structure for learned continuous dynamics. CDLM on the other hand remains *discrete-native*, leveraging exact posterior bridges of the original process. The distinction is therefore not superiority, but modeling choice. We expect these directions to coexist—much like flow and diffusion models in continuous domains—and believe a direct, compute-matched comparison would be valuable future work.
>
> **Diversity metrics [Q2].**
> We will make diversity metrics more explicit in revision. We already report token entropy in main tables and MAUVE scores in appendix. In this conditional setting, MAUVE is largely saturated, while entropy is more discriminative. This is particularly visible for DUO-DCD greedy sampling, where strong perplexity coexists with sharply reduced entropy and degraded conditioning fidelity. We will also clarify this interpretation. Our ablations (Table 8) further show JSD provides the best quality–diversity tradeoff, while backward KL becomes overly mode-seeking.
>
> We hope these clarifications address your concerns and better reflect the scope, generality, and empirical strength of the proposed approach.
>
> ### **References**
> [1] Austin et al., “Structured Denoising Diffusion Models in Discrete State-Spaces,” NeurIPS, 2021
>
> [2] Vignac et al., “DiGress: Discrete Denoising Diffusion for Graph Generation,” ICLR, 2023
>
> [3] Plasser et al., “Discrete Diffusion Probabilistic Models for Symbolic Music Generation,” IJCAI, 2023
>
> [4] Elamvazhuthi et al., “Discrete Diffusion with Sample-Efficient Estimators for Conditionals,” arXiv, 2026

---

> > ### Author Rebuttal · Reviewer_c4BZ · 2026-04-03
> >
> > Appreciate authors detailed feedback, will consider updating rating later.

---

> > > ### Author Response · Authors · 2026-04-07
> > >
> > > Thank you again for the careful and continued engagement with our work. We are glad the clarifications were helpful, and we especially appreciate your openness to revisiting the (weak accept) rating.

---

### Decision · Program_Chairs · 2026-04-30

**Decision:**

Accept (regular)

**Comment:**

This paper proposes Consistent Diffusion Language Models (CDLM), a framework for accelerating discrete diffusion models via a consistency-based training principle. Since discrete diffusion lacks a probability flow ODE, the authors introduce Multi-Path Discrete Consistency (MPDC), which enforces consistency across stochastic posterior bridges between noise levels. This yields a path-independent denoising objective that unifies diffusion, consistency training, and distillation. Empirically, the method demonstrates strong performance on text generation benchmarks, particularly in the few-step regime.

Across all four reviewers, there is clear agreement that the paper is technically sound, clearly written, and empirically strong. A key strength is the ability to achieve high-quality few-step generation, often outperforming both base diffusion models and multi-stage distilled baselines. The formulation is viewed as clean and well-motivated, and the single-stage training approach is seen as practically appealing. Overall, the empirical gains—especially in low-step generation—are considered significant.

The main concerns are also consistent. First, regarding novelty, reviewers note that the individual components are largely known, and the contribution lies in their combination and adaptation to the discrete setting. Second, regarding efficiency, CDLM incurs higher per-step training cost, and comparisons are not fully FLOPs-matched. Third, regarding scope, experiments are limited to the 110M scale, and the method relies on analytic posterior bridges. Finally, one reviewer raises a theoretical limitation: the factorized predictor does not resolve the difficulty of modeling joint token dependencies in the extreme one-step setting, which limits the interpretation of the objective.

Overall, these concerns do not outweigh the strengths. The paper presents a coherent and practically effective framework for improving few-step discrete diffusion, with strong empirical support.